# A Single-Loop Accelerated Extra-Gradient Difference Algorithm with Improved Complexity Bounds for Constrained Minimax Optimization

**Yuanyuan Liu[1], Fanhua Shang[2],\*, Weixin An[1], Junhao Liu[1], Hongying Liu[3,6],\*, Zhouchen Lin[4,5,6]**

[1]Key Laboratory of Intelligent Perception and Image Understanding of Ministry of Education,
School of Artificial Intelligence, Xidian University, China
[2]School of Computer Science and Technology, College of Intelligence and Computing, Tianjin University
[3]Medical College, Tianjin University, China
[4]National Key Lab of General AI, School of Intelligence Science and Technology, Peking University
[5]Institute for Artificial Intelligence, Peking University
[6]Peng Cheng Laboratory
yyliu@xidian.edu.cn, fhshang@tju.edu.cn, hyliu2009@tju.edu.cn
zlin@pku.edu.cn

## Abstract

In this paper, we propose a novel extra-gradient difference acceleration algorithm for solving constrained nonconvex-nonconcave (NC-NC) minimax problems. In particular, we design a new extra-gradient difference step to obtain an important quasi-cocoercivity property, which plays a key role to significantly improve the convergence rate in the constrained NC-NC setting without additional structural assumption. Then momentum acceleration is also introduced into our dual accelerating update step. Moreover, we prove that, to find an $\epsilon$-stationary point of the function $f$, the proposed algorithm attains the complexity $\mathcal{O}(\epsilon^{-2})$ in the constrained NC-NC setting, while the best-known complexity bound is $\widetilde{\mathcal{O}}(\epsilon^{-4})$, where $\widetilde{\mathcal{O}}(\cdot)$ hides logarithmic factors compared to $\mathcal{O}(\cdot)$. As the special cases of the constrained NC-NC setting, the proposed algorithm can also obtain the same complexity $\mathcal{O}(\epsilon^{-2})$ for both the nonconvex-concave (NC-C) and convex-nonconcave (C-NC) cases, while the best-known complexity bounds are $\widetilde{\mathcal{O}}(\epsilon^{-2.5})$ for the NC-C case and $\widetilde{\mathcal{O}}(\epsilon^{-4})$ for the C-NC case. For fair comparison with existing algorithms, we also analyze the complexity bound to find $\epsilon$-stationary point of the primal function $\phi$ for the constrained NC-C problem, which shows that our algorithm can improve the complexity bound from $\widetilde{\mathcal{O}}(\epsilon^{-3})$ to $\mathcal{O}(\epsilon^{-2})$. To the best of our knowledge, this is the first time that the proposed algorithm improves the best-known complexity bounds from $\widetilde{\mathcal{O}}(\epsilon^{-4})$ and $\widetilde{\mathcal{O}}(\epsilon^{-3})$ to $\mathcal{O}(\epsilon^{-2})$ in both the NC-NC and NC-C settings.

## 1 Introduction

This paper considers the following smooth minimax optimization problem,

$$\min_{x\in\mathcal{X}} \max_{y\in\mathcal{Y}} f(x,y), \tag{1}$$

---

*Corresponding authors

37th Conference on Neural Information Processing Systems (NeurIPS 2023).

where $\mathcal{X} \subseteq \mathbb{R}^m$ and $\mathcal{Y} \subseteq \mathbb{R}^n$ are nonempty closed and convex feasible sets, and $f : \mathbb{R}^m \times \mathbb{R}^n \to \mathbb{R}$ is a smooth function. In recent years, this problem has drawn considerable interest from machine learning and other engineering communities such as generative adversarial networks [14], adversarial machine learning [15, 28], game theory [3], reinforcement learning [12, 41], empirical risk minimization [54, 39], and robust optimization [36, 13, 5]. While there is an extensive body of literature on minimax optimization, most prior works such as [54, 39, 42, 33, 50] focus on the convex-concave setting, where $f(x, y)$ is convex in $x$ and concave in $y$. However, in many nonconvex minimax machine learning problems as in [35, 6], $f(x, y)$ is nonconvex in $x$ and (strongly) concave in $y$, or $f(x, y)$ is (strongly) convex in $x$ and nonconcave in $y$. For *nonconvex-strongly-concave* (NC-SC) minimax problems, a number of efficient algorithms such as [34, 25, 27] were proposed, and their complexity can be $\widetilde{\mathcal{O}}(\kappa_y^2 \epsilon^{-2})$ for achieving an $\epsilon$-stationary point $\hat{x}$ (e.g., $\|\nabla \phi(\hat{x})\| \leq \epsilon$) of the primal function $\phi(\cdot) := \max_{y \in \mathcal{Y}} f(\cdot, y)$, where $\kappa_y$ is the condition number for $f(x, \cdot)$, and $\widetilde{\mathcal{O}}$ hides logarithmic factors. More recently, [24] proposed an accelerated algorithm, which improves the gradient complexity to $\widetilde{\mathcal{O}}(\sqrt{\kappa_y}\epsilon^{-2})$, which exactly matches the lower complexity bound in [52, 23]. Therefore, we mainly consider the problem (1) in nonconvex-nonconcave (NC-NC), nonconvex-concave (NC-C) and convex-nonconcave (C-NC) settings.

## 1.1 Algorithms in NC-C and C-NC Settings

For *nonconvex-concave* (NC-C, but not strongly concave) and *convex-nonconcave* (C-NC) problems, there are two types of algorithms (i.e., multi-loop (including double-loop and triple-loop) and single-loop algorithms), and most of them are multi-loop algorithms such as [18, 31, 40, 24]. [31] proposed a multi-step framework that finds an $\epsilon$-first order Nash equilibrium of $f(x, y)$ with the complexity $\widetilde{\mathcal{O}}(\epsilon^{-3.5})$. [40] designed a proximal dual implicit accelerated algorithm and proved that their algorithm finds an $\epsilon$-stationary point of $\phi$ with the complexity $\widetilde{\mathcal{O}}(\epsilon^{-3})$. More recently, [24] proposed an accelerated algorithm, which achieves the complexity $\widetilde{\mathcal{O}}(\epsilon^{-2.5})$ to find an $\epsilon$-stationary point of $f$. These multi-loop algorithms require at least $\mathcal{O}(\epsilon^{-2})$ outer iterations, and thus their complexities are more than $\mathcal{O}(\epsilon^{-2})$. Even though single-loop methods are more popular in practice due to their simplicity, few single-loop algorithms have been proposed for NC-C setting. The most natural approach is the gradient descent-ascent (GDA) method, which performs a gradient descent step on $x$ and a gradient ascent step on $y$ at each iteration. However, GDA fails to converge even for simple bilinear zero-sum games [22]. Subsequently, several improved GDA algorithms such as [8, 25, 27, 44] were proposed. For instance, [25] proved that the complexity of their two-time-scale GDA to find an $\epsilon$-stationary point of $\phi$ is $\mathcal{O}(\epsilon^{-6})$ for NC-C problems. Moreover, [27] presented an efficient algorithm, which obtains an $\epsilon$-stationary point of $f$ with the complexity $\mathcal{O}(\epsilon^{-4})$. In fact, the complexity only counts the number of times the maximization subproblem is solved, and does not consider the complexity of solving this subproblem. [44] proposed a unified algorithm, and proved that the complexity of their algorithm to find an $\epsilon$-stationary point of $f$ is $\mathcal{O}(\epsilon^{-4})$ for both NC-C and C-NC problems. More recently, [51] presented a smoothed-GDA algorithm and proved that the complexity can be improved to $\mathcal{O}(\epsilon^{-2})$ for optimizing a special case of NC-C problems (i.e., minimizing the pointwise maximum of a finite collection of nonconvex functions). However, its complexity is still $\mathcal{O}(\epsilon^{-4})$ for both NC-C and C-NC problems. One natural question is: can we design a single-loop accelerated algorithm with the optimal complexity bound $\mathcal{O}(\epsilon^{-2})$ for *both NC-C and C-NC problems*?

## 1.2 Algorithms in NC-NC Settings

This paper mainly considers constrained NC-NC minimax problems, i.e., $f(x, y)$ is nonconvex in $x$ and nonconcave in $y$ in the constrained setting (called constrained NC-NC). In recent years, some works such as [7, 31, 46, 38, 20] focus on structured NC-NC problems. That is, the saddle gradient operator of such minimax problems or their objectives must satisfy one of the structured assumptions: the minty variational inequality (MVI) condition, the weak MVI condition, or negative comonotone condition and the Polyak-Łojasiewicz condition. However, structured NC-NC problems have limited practical applications because they are required to satisfy strong structural assumptions. For more practical constrained NC-NC problems or more general NC-NC problems, the convergence guarantee of the algorithms is still a challenge. In this paper, we also focus on convergence analysis

Table 1: Comparison of complexities of the minimax algorithms to find an $\epsilon$-stationary point of $f(\cdot, \cdot)$ in the NC-C, C-NC and NC-NC settings. Note that Smoothed-GDA [51] can find an $\epsilon$-stationary point for a special problem of (1) with $\mathcal{O}(\epsilon^{-2})$, and $\widetilde{\mathcal{O}}$ hides logarithmic factors compared to $\mathcal{O}(\cdot)$.

| Optimality Criteria | References | NC-C | C-NC | NC-NC | Simplicity |
|---|---|---|---|---|---|
| Stationarity of $f$ with smoothness and compact sets assumptions | Lu et al. [27] | $\widetilde{\mathcal{O}}(\epsilon^{-4})$ | - | - | Single-Loop |
| | Nouiehed et al. [31] | $\widetilde{\mathcal{O}}(\epsilon^{-3.5})$ | - | - | Multi-Loop |
| | Lin et al. [24] | $\widetilde{\mathcal{O}}(\epsilon^{-2.5})$ | - | - | Multi-Loop |
| | Zhang et al. [51] | $\mathcal{O}(\epsilon^{-4})$ | - | - | Single-Loop |
| | Xu et al. [44] | $\mathcal{O}(\epsilon^{-4})$ | $\mathcal{O}(\epsilon^{-4})$ | - | Single-Loop |
| | **This work** (Theorem 1) | $\mathcal{O}(\epsilon^{-2})$ | $\mathcal{O}(\epsilon^{-2})$ | $\mathcal{O}(\epsilon^{-2})$ | Single-Loop |

Table 2: Comparison of complexities of existing minimax algorithms and the proposed algorithm to find an $\phi(\cdot) := \max_{y \in \mathcal{Y}} f(\cdot, y)$ in the nonconvex-concave (NC-C) setting. This table only highlights the dependence of $\epsilon$, and compared with $\mathcal{O}(\cdot)$, $\widetilde{\mathcal{O}}(\cdot)$ hides logarithmic factors.

| NC-C Settings | References | Compact set | Complexity | Simplicity |
|---|---|---|---|---|
| NC-C (Stationarity of $\phi$) | Rafique et al. [34], Jin et al. [17] | $\mathcal{X}, \mathcal{Y}$ | $\widetilde{\mathcal{O}}(\epsilon^{-6})$ | Multi-Loop |
| | Lin et al. [25] | $\mathcal{X}, \mathcal{Y}$ | $\widetilde{\mathcal{O}}(\epsilon^{-6})$ | Single-Loop |
| | Thekumprampil et al. [40] | $\mathcal{X}, \mathcal{Y}$ | $\widetilde{\mathcal{O}}(\epsilon^{-3})$ | Multi-Loop |
| | Zhao [55], Lin et al. [24] | $\mathcal{X}, \mathcal{Y}$ | $\widetilde{\mathcal{O}}(\epsilon^{-3})$ | Multi-Loop |
| | **This work** (Theorem 2) | $\mathcal{Y}$ | $\mathcal{O}(\epsilon^{-2})$ | Single-Loop |

for solving more practical constrained NC-NC problems. *Another natural question is: can we design a single-loop accelerated algorithm to further improve the bound in the constrained NC-NC setting?*

### 1.3 Motivations and Our Contributions

**Motivations:** For NC-C minimax problems, can we design a single-loop directly accelerated algorithm with the gradient complexity lower than the best-known result $\widetilde{\mathcal{O}}(\epsilon^{-2.5})$? Though Smoothed-GDA [51] can obtain the complexity $\mathcal{O}(\epsilon^{-2})$ for a special case of Problem (1), it only attains the complexity $\mathcal{O}(\epsilon^{-4})$ for NC-C minimax problems. For C-NC minimax problems, for any given $x$, to solve the nonconcave maximization subproblem with respect to $y$ is NP-hard. As a result, all existing multi-loop algorithms will lose their theoretical guarantees as discussed in [44]. Can we propose a single-loop directly accelerated algorithm with the complexity lower than the best-known result $\mathcal{O}(\epsilon^{-4})$ for C-NC and NC-NC minimax problems?

**Our Contributions:** This paper proposes a novel single-loop extra-gradient difference acceleration algorithm to push towards optimal gradient complexities for constrained NC-NC minimax problems (1), and answer the above-mentioned problems.

We summarize the major contributions of this paper.

• We design a new single-loop accelerating algorithm for solving *constrained NC-NC* problems. In the proposed algorithm, we design a new extra-gradient difference scheme, and combine the gradient ascent and momentum acceleration steps for the dual variable update. In our algorithm, we present an important quasi-cocoercivity property. By leveraging the quasi-cocoercivity property, we can improve the complexity bound in our theoretical analysis.

• We analyze the convergence properties of the proposed algorithm for *constrained NC-NC* problems. Theorem 1 shows that to find an $\epsilon$-stationary point of $f$, the proposed algorithm can obtain the gradient complexity $\mathcal{O}(\epsilon^{-2})$, which is the first time to attains the complexity bound in constrained NC-NC setting. The *constrained NC-C and C-NC* problems can be viewed as two special cases of the *constrained NC-NC* problem, and the proposed algorithm is also applicable to these two special problems. And its complexity is still $\mathcal{O}(\epsilon^{-2})$ for both NC-C and C-NC problems, which significantly improves the gradient complexity from $\mathcal{O}(\epsilon^{-4})$ of existing single-loop algorithms or $\widetilde{\mathcal{O}}(\epsilon^{-2.5})$ of existing multi-loop algorithms to $\mathcal{O}(\epsilon^{-2})$. The complexities of some recently proposed algorithms are listed in Table 1.

• In order to make a comprehensive comparison with existing algorithms, we also provide the theoretical analysis of our algorithm in terms of another convergence criteria (i.e., an $\epsilon$-stationary point of $\phi$) for constrained NC-C problems. The result shows that our algorithm improves the best-known result as in [40, 24, 55] from $\widetilde{\mathcal{O}}(\epsilon^{-3})$ to $\mathcal{O}(\epsilon^{-2})$, as shown in Table 2.

## 2 Preliminaries and Related Works

**Notation:** Throughout this paper, we use lower-case letters to denote vectors such as $x, y$, and calligraphic upper-case letters to denote sets such as $\mathcal{X}, \mathcal{Y}$. For a differentiable function $f$, $\nabla f(x)$ is the gradient of $f$ at $x$. For a function $f(\cdot, \cdot)$ of two variables, $\nabla_x f(x, y)$ (or $\nabla_y f(x, y)$) is the partial gradient of $f$ with respect to the first variable (or the second variable) at $(x, y)$. For a vector $x$, $\|x\|$ denotes its $\ell_2$-norm. We use $\mathcal{P}_{\mathcal{X}}$ and $\mathcal{P}_{\mathcal{Y}}$ to denote projections onto the sets $\mathcal{X}$ and $\mathcal{Y}$.

**Assumption 1** (Smoothness). *$f(\cdot, \cdot)$ is continuously differentiable, and there exists a positive constant $L$ such that*

$$\|\nabla_x f(x_1, y_1) - \nabla_x f(x_2, y_2)\| \leq L\|x_1 - x_2\|, \ \ \|\nabla_y f(x_1, y_1) - \nabla_y f(x_2, y_2)\| \leq L\|y_1 - y_2\|$$

*holds for all $x_1, x_2 \in \mathbb{R}^m$, $y_1, y_2 \in \mathbb{R}^n$.*

**Definitions of the monotone operators:** A operator $F(\cdot) : \mathbb{R}^n \to \mathbb{R}^n$ is monotone, if $[F(s) - F(t)]^T(s - t) \geq 0$, $\forall s, t \in \mathbb{R}^n$. If $[F(s) - F(t)]^T(s - t) \leq 0$, $F$ is negative monotone.

A mapping $F(\cdot)$ is co-coercive if there is a positive constant $\alpha$, such that

$$[F(s) - F(t)]^T(s - t) \geq \alpha\|F(s) - F(t)\|^2, \ \forall s, t \in \mathbb{R}^n.$$

**Nonconvex-Concave Minimax Optimization:** Due to the nonconvex nature of these minimax problems, finding the global solution is NP-hard in general. The recently proposed algorithms aim to find stationary solutions to such problems. For instance, the first-order Nash equilibrium condition (called game stationary) is used as an optimality condition in [31]. Besides game stationary, there are two main optimality criteria (i.e., an $\epsilon$-stationary point of $f(\cdot, \cdot)$ or $\phi(\cdot) := \max_{y \in \mathcal{Y}} f(\cdot, y)$) for the convergence analysis of the algorithms such as [4, 44, 24].

For solving NC-C minimax problems, there exist a number of efficient multi-loop and single-loop algorithms such as [31, 25, 4, 44, 24]. Most of them are multi-loop algorithms, which either employ an accelerated update rule of $x$ by adding regularization terms to its subproblem, or use multiple gradient ascent steps for the update of $y$ to solve the subproblem exactly or inexactly. Compared with multi-loop algorithms, single-loop algorithms are easier to implement. One of the most popular single-loop algorithms is GDA. However, GDA with a constant stepsize can fail to converge even for a simple bilinear minimax problem [29].

To address this issue, only a few single-loop algorithms such as [25, 27, 44] were proposed, and most of them employed a smoothing or proximal point technique. For instance, Smoothed-GDA [51] introduces a smooth function $\varphi(x, y, z) = f(x, y) + \frac{a}{2}\|x - z\|^2$ for the update of the primal variable $x$, where $z$ is an auxiliary variable and $a$ is a constant, and its main update steps are

$$x_{t+1} = \mathcal{P}_{\mathcal{X}}(x_t - \eta_x \nabla_x \varphi(x_t, z_t, y_t)), \ y_{t+1} = \mathcal{P}_{\mathcal{Y}}(y_t + \eta_y \nabla_y f(x_{t+1}, y_t)), \ z_{t+1} = z_t + \beta(x_{t+1} - z_t),$$

where $\eta_x, \eta_y > 0$ are two stepsizes, and $0 < \beta \leq 1$. Smoothed-GDA can obtain the gradient complexity, $\mathcal{O}(\epsilon^{-4})$, for nonconvex-concave minimax problems.

**Extra-Gradient Methods:** There are some extra-gradient methods such as [48] for solving minimax problems. Let $\eta_0 > 0$ and $u_t$ be the $t$-th iterate, the extra-gradient method [53] has the following projection-type prediction-correction step:

$$\text{Prediction} : u_{t+1/2} = \mathcal{P}_{\Omega}(u_t - \eta_t F(u_t)), \ \text{Correction} : u_{t+1} = \mathcal{P}_{\Omega}(u_t - \eta_t F(u_{t+1/2})).$$

In this paper, we call $u_{t+1/2}$ as the prediction point at the $t$-iteration.

## 3 Single-Loop Extra-Gradient Difference Acceleration Algorithm

In this section, we propose a single-loop Extra-Gradient Difference Acceleration algorithm (EGDA) for solving constrained NC-NC minimax problems.

---
**Algorithm 1** EGDA for NC-NC minimax problems
---
1: **Initialize:** $x_0, y_0, u_0, u_{-1/2}, \tau > 0.5, \eta_x, \eta_y^t, \beta$.
2: **for** $t = 0, 1, \ldots, T - 1$ **do**
3:    $x_{t+1} = \mathcal{P}_{\mathcal{X}}[x_t - \eta_x \nabla_x f(x_t, y_t)];$        $u_{t+1/2} = y_t + \beta[\nabla_y f(x_t, u_{t-1/2}) - \nabla_y f(x_t, y_{t-1})];$
4:    $u_{t+1} = \mathcal{P}_{\mathcal{Y}}\left[y_t + \eta_y^t \nabla_y f(x_t, u_{t-1/2})\right];$        $y_{t+1} = \tau y_t + (1 - \tau) u_{t+1};$
5: **end for**
6: **Output:** $(x_T, y_T)$.
---

## 3.1 Extra-Gradient Difference Acceleration

In recent years, many algorithms such as [27, 31, 32, 24, 51, 44] have been proposed for solving NC-C minimax problems. In essence, most of them are "indirect" acceleration algorithms, which are used to optimize the surrogate functions with a smoothing or proximal point term instead of the original function. However, this may hurt the performance of these algorithms both in theory and in practice [2, 1]. To address this issue, we propose a single-loop directly accelerating algorithm to find an $\epsilon$-stationary points of $f$ and $\phi$ with a significantly improved complexity $\mathcal{O}(\epsilon^{-2})$. The main update steps of our EGDA algorithm are designed as follows.

- Gradient descent:

$$x_{t+1} = \arg\min_{x \in \mathcal{X}} \left\{ \langle \nabla_x f(x_t, y_t), \, x \rangle + \|x - x_t\|^2 / \eta_x \right\}. \tag{2}$$

- Extra-gradient difference prediction:

$$u_{t+1/2} = y_t + \beta[\nabla_y f(x_t, u_{t-1/2}) - \nabla_y f(x_t, y_{t-1})]. \tag{3}$$

- Gradient ascent correction:

$$u_{t+1} = \arg\max_{u \in \mathcal{Y}} \left\{ \langle \nabla_y f(x_t, u_{t-1/2}), \, u \rangle - \|u - y_t\|^2 / \eta_y^t \right\}. \tag{4}$$

- Momentum acceleration:

$$y_{t+1} = \tau y_t + (1 - \tau) u_{t+1}. \tag{5}$$

Here, $0 < \beta < \frac{1}{L}$, $\eta_x, \eta_y^t > 0$ are two stepsizes, $u_{t+1/2}, u_{t+1}$ are auxiliary variables, $u_{t+1/2}$ is a prediction point, and $1/2 < \tau \leq 1$ is a momentum parameter. Our EGDA algorithm is formally presented in Algorithm 1. Our EGDA algorithm first performs one proximal gradient descent step on the primal variable $x$, and then we design a new dual-accelerating scheme in (3), (4) and (5) for the dual variable $y$.

## 3.2 Advantages of Our Algorithm and Comparison to Related Work

We first design a new prediction point $u_{t+1/2}$ in Eq. (3). Compared with extra-gradient-type methods such as [7, 16, 20], one of the main differences is that the proposed prediction point $u_{t+1/2}$ in (3) is updated by the gradient difference (i.e., $\nabla_y f(x_t, u_{t-1/2}) - \nabla_y f(x_t, y_{t-1})$), while the prediction point in other extra-gradient-type algorithms is updated only by using the gradient information at the correction point $u_t$. Then the gradient at the new prediction point $u_{t+1/2}$ is used in the gradient ascent step (4). And the dual variable $y$ is update by the momentum acceleration step in (5).

- **Prediction Point:** The monotonicity and co-coercivity properties of gradient operators play a crucial role for convergence analysis. However, these important properties do not always hold for nonconvex problems. Some researchers have made great progress in some special nonconvex settings, such as structured nonconvex and weakly convex settings, which require a weaker condition such as weakly monotone [26], pseudo-monotone [16], and MVI [7]. However, such conditions seriously limit the application scope of Problem (1).

To address this challenge, we design a new prediction point scheme in (3), which can help us obtain a useful quasi-cocoercivity property. As a result, it does not require any monotone or structural assumption. Specifically, we find that we only require a weaker property in our theoretical analysis, that is, the co-coercivity is required at some special points $\{u_{t+1/2}, y_t\}$ ($\langle \nabla_y f(x_t, u_{t+1/2}) - \nabla_y f(x_t, y_t), u_{t+1/2} - y_t \rangle \geq \beta \|\nabla_y f(x_t, u_{t+1/2}) - \nabla_y f(x_t, y_t)\|^2$ with $\beta > 0$). Thus, we develop a

decoupling idea to construct the prediction point $u_{t+1/2}$. That is, we use the gradients with respect to $y$ at $u_{t-1/2}, y_{t-1}$ instead of those at the points $u_{t+1/2}, y_t$. We can obtain a property (called the quasi-co-coercivity) in Section 4.2, which plays a key role in our theoretical analysis.

• **Gradient Difference:** We also briefly discuss the underlying intuition of the proposed gradient difference in (3). We find that the proposed update in (3) is similar to the forms in [43] (see Eqs. (12) and (14) in [43]). [43] proposed a first-order procedure (i.e., difference of gradients) to achieve the negative curvature of a Hessian matrix. Therefore, our algorithm has a similar procedure, which contains second-order information. Moreover, we use the difference of gradients in the gradient ascent procedure for the dual update. But our EGDA algorithm only requires the Lipschitz gradient assumption for minimax problems to find first-order stationary points, while [43] requires both the Lipschitz gradient and Lipschitz Hessian assumptions for solving second-order stationary points of nonconvex minimization problems.

• **Momentum Acceleration:** We design a dual-accelerating update rule in (4) for the dual variable $y$ in our EGDA algorithm, which is different from standard momentum acceleration schemes as in [31, 40, 24]. That is, the accelerated rules of existing algorithms are for the primal variable $x$, while our accelerated rule is designed for the dual variable $y$.

Therefore, the proposed new dual-accelerating step (including the gradient different prediction step in (3), the gradient ascent correction in (4) and momentum acceleration in (5)) is a key accelerated technique for our EGDA algorithm, and can help to improve the complexity bound from $\mathcal{O}(\epsilon^{-4})$ to $\mathcal{O}(\epsilon^{-2})$. In particular, our EGDA algorithm performs both gradient descent and ascent steps to the original function $f$. In contrast, many existing algorithms such as [51, 47, 44, 18] optimize their surrogate functions with smoothing terms instead of the original function. In particular, their smoothing parameters need to tune by repeatedly executing the algorithms, which may make them impractical [1]. As in our theoretical guarantees below, the proposed single-loop algorithm is able to significantly improve the best-known gradient complexity, $\mathcal{O}(\epsilon^{-4})$, of existing single-loop algorithms such as [27, 51, 44] to $\mathcal{O}(\epsilon^{-2})$.

## 4 Convergence Guarantees

In this section, we provide the convergence guarantees of our EGDA algorithm (i.e., Algorithm 1) for solving constrained NC-NC and NC-C problems. We first present the definitions of the two optimality criteria (i.e., an $\epsilon$-stationary point of $f$ or $\phi$). All the proofs of the lemmas, properties and theorems below are included in the Supplementary Material.

### 4.1 Optimality Criteria and Key Property

Since finding a global minimum of a nonconvex optimization problem is NP-hard, finding a global saddle point (or Nash equilibrium) of a NC-NC function $f$ is intractable [30]. As in the literature in the *NC-NC* setting, we introduce the local surrogates (i.e., the stationary point of $f$) and in the *NC-C setting*, we introduce the local surrogates (i.e., the stationary point of $f$ or $\phi$), whose gradient mappings are equal to zero. Below we define the following two optimality measures (i.e., an $\epsilon$-stationary points of $f$ or $\phi$) for our theoretical analysis.

**Definition 1** (An $\epsilon$-stationary point of $f$ [27]). *A point $(\overline{x}, \overline{y}) \in \mathcal{X} \times \mathcal{Y}$ is an $\epsilon$-stationary point of $f(\cdot, \cdot)$ if $\|\pi(\overline{x}, \overline{y})\| \leq \epsilon$, where $\eta_x$ and $\eta_y$ are two constants, and*

$$\pi(\overline{x}, \overline{y}) := \begin{bmatrix} (1/\eta_x)(\overline{x} - \mathcal{P}_{\mathcal{X}}(\overline{x} - \eta_x \nabla_x f(\overline{x}, \overline{y}))) \\ (1/\eta_y)(\overline{y} - \mathcal{P}_{\mathcal{Y}}(\overline{y} + \eta_y \nabla_y f(\overline{x}, \overline{y}))) \end{bmatrix}. \tag{6}$$

*If $\epsilon = 0$, then $(\overline{x}, \overline{y})$ is a stationary point of $f$.*

For the *NC-C* setting, we also present another convergence criterion used in [40, 24]. Let $\phi(x) := \max_{y \in \mathcal{Y}} f(x, y)$, $\hat{x}$ is called an $\epsilon$-stationary point of a smooth primal function $\phi : \mathbb{R}^m \to \mathbb{R}$, if $\|\nabla \phi(\hat{x})\| \leq \epsilon$. However, the function $\phi$ is not necessarily differentiable for minimax problems. Following [9, 40, 24], we introduce the Moreau envelope of $\phi$ for the optimality criterion, especially when $\phi$ is a weakly convex function, i.e., $\phi$ is $L$-weakly convex if the function $\phi(\cdot) + \frac{L}{2}\|\cdot\|^2$ is convex. We refer readers to [9, 24] for the comparison of these two criteria.

**Definition 2** (An $\epsilon$-stationary point of $L$-weakly convex function $\phi$). *$\hat{x}$ is an $\epsilon$-stationary point of an $L$-weakly convex function $\phi : \mathbb{R}^m \to \mathbb{R}$, if $\|\nabla\phi_{1/(2L)}(\hat{x})\| \leq \epsilon$, where $\phi_{1/(2L)}$ is the Moreau envelope of $\phi$ and is defined as: $\phi_\rho(x) := \min_z \phi(z) + (1/2\rho)\|z-x\|^2$. If $\epsilon = 0$, then $\hat{x}$ is a stationary point.*

Then we give the following important property for the analysis of the proposed algorithm. By leveraging the property, we can obtain the complexity bound of the proposed algorithm.

**Property 1** (**Quasi-Cocoercivity**). *Let $u_{t+1/2}$ be updated in Eq. (3), then*

$$\langle \nabla_y f(x_t, u_{t-1/2}) - \nabla_y f(x_t, y_{t-1}), \ u_{t+1/2} - y_t \rangle = \beta\|\nabla_y f(x_t, u_{t-1/2}) - \nabla_y f(x_t, y_{t-1})\|^2.$$

## 4.2  Core Lemma

Our theoretical results mainly rely on Lemma 1 below, which plays a key role in the proofs of Theorems 1 and 2. Let $\{(x_t, y_t, u_t, u_{t-1/2})\}$ be a sequence generated by Algorithm 1, and we define the potential function

$$G_t := f(x_t, y_t) + 9L\|u_t - y_{t-1}\|^2 + 8L\beta^2\|\nabla_y f(x_{t-1}, u_{t-3/2}) - \nabla_y f(x_{t-1}, y_{t-2})\|^2.$$

Next, we need to prove that our defined potential function can make sufficient decrease at each iteration, i.e., $G_t - G_{t+1} > 0$ as in Lemma 1 below. To prove Lemma 1, we will provide and prove the following upper bounds.

**Proposition 1** (Upper bound of primal-dual updates). *Suppose Assumption 1 holds. Let $\{(x_t, y_t, u_t)\}$ be a sequence generated by Algorithm 1 with $\eta_y^t = \min\{\frac{\beta\|\nabla_y f(x_t, u_{t-1/2}) - \nabla_y f(x_t, y_{t-1})\|^2}{2\|\nabla_y f(x_t, u_{t-1/2})\|^2}, \frac{1}{28L}, \eta_x\}$.*

$$f(x_{t+1}, u_{t+1/2}) - f(x_t, y_t)$$
$$\leq -\left(\frac{1}{\eta_x} - \frac{L}{2}\right)\|x_{t+1} - x_t\|^2 + L\|u_{t+1/2} - y_t\|^2 + L\|y_t - y_{t-1}\|^2 + \underbrace{\langle \nabla_y f(x_t, y_{t-1}), u_{t+1/2} - y_t \rangle}_{A_1}.$$

**Proposition 2** (Upper bound of dual updates). *Suppose Assumption 1 holds. Let $\{(x_t, y_t, u_t)\}$ be a sequence generated by Algorithm 1, then*

$$f(x_{t+1}, y_{t+1}) - f(x_{t+1}, u_{t+1/2})$$
$$\leq \underbrace{\tau\langle \nabla_y f(x_t, u_{t-1/2}), y_t - u_{t+1}\rangle + \langle \nabla_y f(x_t, u_{t-1/2}), u_{t+1} - u_{t+1/2}\rangle}_{A_2 \qquad\qquad\qquad\qquad A_3} + a_t,$$

*where $a_t := 2L\|x_{t+1} - x_t\|^2 + 6L\beta^2\|\nabla_y f(x_t, u_{t-1/2}) - \nabla_y f(x_t, y_{t-1})\|^2 + 8L(1-\tau)^2\|u_t - y_{t-1}\|^2 + 8L\beta^2\|\nabla_y f(x_{t-1}, u_{t-3/2}) - \nabla_y f(x_{t-1}, y_{t-2})\|^2 + 3L\|u_{t+1} - y_t\|^2$.*

Using the optimal condition of Problem (4) and our quasi-cocoercivity in Property 1, we can further bound $A_1 + A_2 + A_3$. The proof sketch of Lemma 1 is listed as follows:

$$G_{t+1} - G_t := \underbrace{f(x_{t+1}, y_{t+1}) - f(x_{t+1}, u_{t+1/2})}_{\text{Proposition 1}} + \underbrace{f(x_{t+1}, u_{t+1/2}) - f(x_t, y_t)}_{\text{Proposition 2}} + \text{other terms}$$

$$= \underbrace{A_1 + A_2 + A_3}_{\text{Quasi-Cocoercivity in Property 1}} + \quad \text{other terms}.$$

Combining the above results and the definition of $G_t$, we can get the descent estimate of the potential function in Lemma 1, which is a main lemma for Theorems 1 and 2 below.

**Lemma 1** (Descent estimate of $G$). *Suppose Assumption 1 holds. Let $\{(x_t, y_t, u_t)\}$ be a sequence generated by Algorithm 1, then*

$$G_0 - G_T$$
$$= \sum_{t=0}^{T-1}(G_t - G_{t+1})$$
$$\geq \sum_{t=0}^{T-1}\left(\frac{1}{4\eta_y^t}\|u_{t+1} - y_t\|^2 + \frac{\beta - 30L\beta^2}{2}\|\nabla_y f(x_t, u_{t-1/2}) - \nabla_y f(x_t, y_{t-1})\|^2 + \frac{1}{2\eta_x}\|x_{t+1} - x_t\|^2\right).$$

## 4.3 Convergence Results

In this subsection, by using the optimality measure in Definitions 1 and 2, we present the following theoretical results in Theorem 1 and Theorem 2 for the constrained NC-NC setting and NC-C setting, respectively.

**Assumption 2.** *$\mathcal{Y}$ is a closed, convex and compact set with a diameter $D_{\mathcal{Y}} > 0$, and $\mathcal{X}$ is a closed and convex set.*

Furthermore, using Lemma 1 and the optimality measure in Definition 1, we will study the relation between $\pi(x_t, u_t)$ and the difference (i.e., $G_t - G_{t+1}$) to obtain the gradient complexity in Theorem 1 by computing the number of iterations to achieve an $\epsilon$-stationary point of $f$.

**Theorem 1** (Stationarity of $f$ in constrained NC-NC settings). *Suppose Assumptions 1 and 2 hold. Let the two stepsizes $\eta_x \leq \frac{1}{4L}$, $\eta_y^t = \min\{\frac{\beta\|\nabla_y f(x_t, u_{t-1/2}) - \nabla_y f(x_t, y_{t-1})\|^2}{2\|\nabla_y f(x_t, u_{t-1/2})\|^2}, \frac{1}{28L}, \eta_x\}$, $\beta \leq \frac{1}{60L}$ and $\tau \geq 1/2$, then the complexity of Algorithm 1 to find an $\epsilon$-stationary point of $f$ is bounded by*

$$\mathcal{O}\Big(\frac{G_0 - \underline{G} + 2LD_{\mathcal{Y}}^2}{\epsilon^2}\Big),$$

*where $G_0 := G(x_0, y_0)$, and $\underline{G} := \min_{x \in \mathcal{X}} \phi(x)$.*

**Remark 1.** *For constrained NC-NC problems, the gradient complexity of our EGDA algorithm to find an $\epsilon$-stationary point of $f$ is $\mathcal{O}(\epsilon^{-2})$. That is, our EGDA algorithm is first to obtain the gradient complexity in constrained NC-NC setting. In addition, our method can achieve the same complexity, $\mathcal{O}(\epsilon^{-2})$, as the algorithm with an additional structured assumption for NC-NC problems. However, different from existing algorithms, our algorithm is more practical. That is, it only requires that $\mathcal{Y}$ is a compact set, while existing algorithms need some stronger structured assumptions in the structured NC-NC setting. The detailed comparison is shown in Table 1. As two special cases of the constrained NC-NC problem, this theoretical result in Theorem 1 can be extended to the constrained NC-C and C-NC settings. For NC-C problems, the gradient complexity of our EGDA algorithm to find an $\epsilon$-stationary point of $f$ is $\mathcal{O}(\epsilon^{-2})$, while the best-known result of single-loop algorithms such as [27, 51, 44] is $\mathcal{O}(\epsilon^{-4})$, and the best-known result of multi-loop algorithms such as [24] is $\widetilde{\mathcal{O}}(\epsilon^{-2.5})$. That is, our EGDA algorithm improves the best-known gradient complexity from $\widetilde{\mathcal{O}}(\epsilon^{-2.5})$ to $\mathcal{O}(\epsilon^{-2})$. Smoothed-GDA [51] can also obtain the complexity $\mathcal{O}(\epsilon^{-2})$ for a special case of Problem (1) and $\mathcal{O}(\epsilon^{-4})$ for general NC-C minimax problems, while our algorithm attains the optimal result $\mathcal{O}(\epsilon^{-2})$ for all NC-C minimax problems. Existing algorithms such as HiBSA [27] and AGP [44] require the compactness of the domain $\mathcal{X}$ in the NC-C setting, while our EGDA algorithm does not, which significantly extends its applicability. For C-NC minimax problems, the proposed algorithm can obtain the complexity, $\mathcal{O}(\epsilon^{-2})$, while the best-known complexity as in [44] is $\mathcal{O}(\epsilon^{-4})$. In other words, the proposed algorithm can improve the best-known result from $\mathcal{O}(\epsilon^{-4})$ to $\mathcal{O}(\epsilon^{-2})$.*

By using the criterion in Definition 2 as the optimality measure, we use the definition of $\phi_{1/2L}$ and introduce the property of $\nabla\phi_{1/2L}$ as in [24]. With a similar setting for varying stepsizes as in [24], we study the relation between $\nabla\phi_{1/2L}$ and the basic descent estimation of the potential function in Lemma 1 by using the property of $\nabla\phi_{1/2L}$, and compute the number of iterations to achieve an $\epsilon$-stationary point of $\phi$. We provide the following theoretical result and its detailed proof in the Supplementary Material.

**Theorem 2** (Stationarity of $\phi$ for constrained NC-C settings). *Using the same notation as in Theorem 1, and $f$ is concave with respect to $y$. Let $\{(x_t, y_t, u_t)\}$ be a sequence generated by Algorithm 1 with the stepsizes $\eta_y^t = \min\{\frac{\beta a_t^2}{2b_t^2}, \frac{1}{28L}, \frac{\beta a_t^2}{2b_t}, \frac{\beta c_t^2}{2d_t}\}$, where $a_t := \|\nabla_y f(x_t, u_{t-1/2}) - \nabla_y f(x_t, y_{t-1})\|, b_t := \|\nabla_y f(x_t, u_{t-1/2})\|, c_t := \|\nabla_y f(x_t, u_{t-1/2})\|, d_t := \|\nabla_y f(x_t, u_t) - \nabla_y f(x_t, u_{t-1/2})\|$. Then the gradient complexity of Algorithm 1 to find an $\epsilon$-stationary point of $\phi$ is bounded by*

$$\mathcal{O}\Big(\frac{D_{\mathcal{Y}}(G_0 - \underline{G} + 2LD_{\mathcal{Y}}^2)}{\epsilon^2}\Big).$$

**Remark 2.** *From Theorem 2, it is clear that the gradient complexity of the proposed algorithm is $\mathcal{O}(\epsilon^{-2})$. That is, Algorithm 1 can improve the best-known gradient complexity from $\widetilde{\mathcal{O}}(\epsilon^{-3})$ as in [24] to $\mathcal{O}(\epsilon^{-2})$. Therefore, Algorithm 1 is the first algorithm, which attains the gradient complexity $\mathcal{O}(\epsilon^{-2})$ to find an $\epsilon$-stationary point of $\phi$ for NC-C minimax problems.*

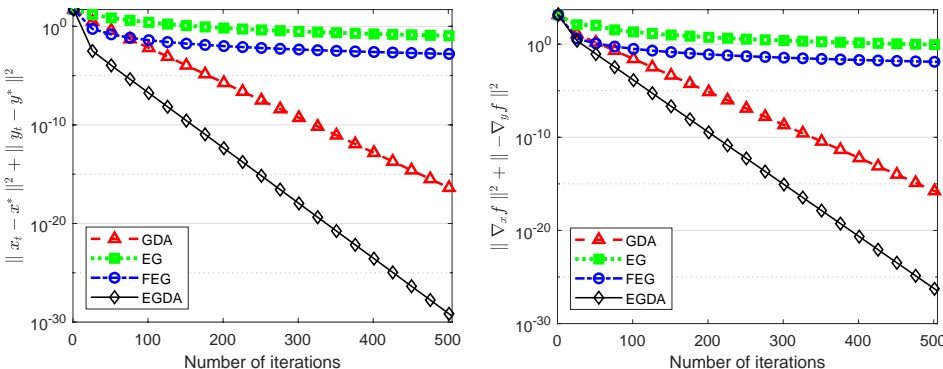

Figure 1: Comparison of all the methods for solving the NC-NC problem, $f(x, y) = x^2 + 3\sin^2 x \sin^2 y - 4y^2 - 10\sin^2 y$. Left: Convergence in terms of $\|x_t - x^*\|^2 + \|y_t - y^*\|^2$, where $(x^*, y^*)$ is the global saddle point; Right: Convergence in terms of $\|\nabla_x f\|^2 + \|\nabla_y f\|^2$.

## 5 Numerical Results

We conduct many experiments to illustrate the performance of the proposed algorithm for solving NC-NC, NC-C and convex-concave problems.

**NC-NC Problems.** We conduct some experiments to illustrate the performance of the proposed algorithm, EGDA, for solving NC-NC problems. Moreover, we compare it against existing methods such as GDA [46], EG [10], EAG [49] and FEG [20]. Detailed setup is provided in the Supplementary Material.

We compare EGDA with GDA, EG and FEG for solving a simple NC-NC minimax problem (i.e., $f(x, y) = x^2 + 3\sin^2 x \sin^2 y - 4y^2 - 10\sin^2 y$), which satisfies the Polyak-Łojasiewicz condition, as shown in Fig. 1. All the results show that EGDA indeed converges to the global saddle point and is significantly faster than other methods including GDA and FEG in terms of both $\|x_t - x^*\|^2 + \|y_t - y^*\|^2$ and $\|\nabla_x f\|^2 + \|\nabla_y f\|^2$. Although FEG has a fast theoretical rate, $\mathcal{O}(1/t^2)$, with a negative monotone assumption, it converges much slower than EGDA in practice.

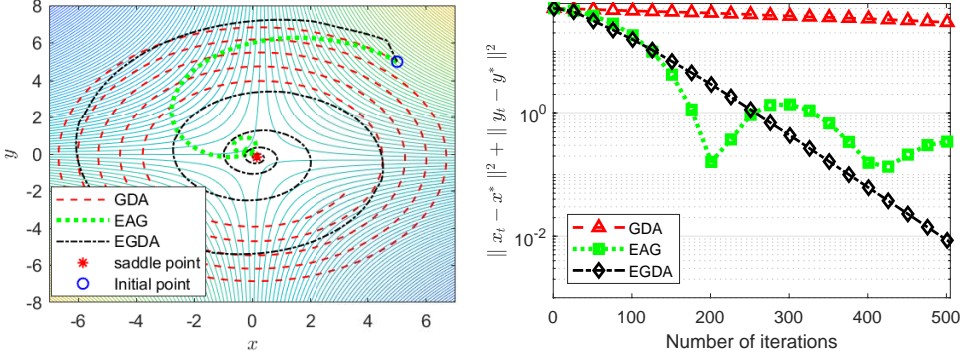

Figure 2: Comparison of all the methods for solving the convex-concave problem, $f(x, y) = \log(1 + e^x) + 3xy - \log(1 + e^y)$. Left: Trajectories of the three algorithms; Right: Convergence in terms of $\|x_t - x^*\|^2 + \|y_t - y^*\|^2$.

**Convex-Concave Problems.** Fig. 2 shows the convergence results of the methods including GDA, EAG [49] and EGDA for solving the convex-concave problem, $f(x, y) = \log(1 + e^x) + 3xy - \log(1 + e^y)$. It is clear that EGDA converges much faster than other methods such as EAG. We also observe empirically when the same step-size is used, even if small, GDA may not converge to stationary points, and it is proven to always have bounded iterates.

**NC-C Problems.** We also apply our EGDA algorithm to train robust neural networks against adversarial attacks on Fashion MNIST and MNIST, and verify our theoretical results. In [15, 28, 31],

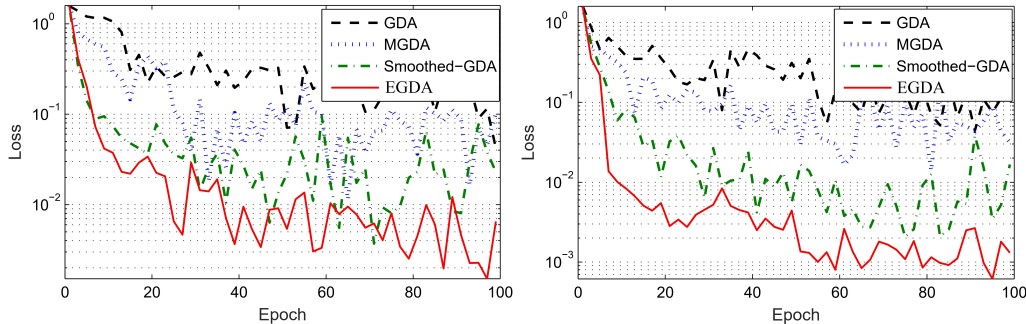

Figure 3: Convergence speed of all the algorithms on Fashion MNIST (left) and MNIST (right).

the robust training procedure can be formulated into a NC-C minimax optimization problem. It is clear that the minimax problem is nonconvex in $x$, but concave in $y$.

FGSM [15] and PGD [19] are two popular methods for generating adversarial examples. To obtain the targeted attack $\hat{x}_{ij}$, we use the same procedure as in [31, 51] as follows. The perturbation level $\varepsilon$ is chosen from $\{0.0, 0.1, 0.2, 0.3, 0.4\}$, and the stepsize is $0.01$. Note that the number of iterations is set to $40$ for FGSM and $10$ for PGD, respectively. The details of the network are included in the Supplementary Material. We compare our EGDA method with GDA [25], MGDA [31] and Smoothed-GDA [51], and illustrate the convergence of all the algorithms on the loss function in Fig. 3. The results show that Smoothed-GDA and EGDA converge significantly faster than other algorithms. This verifies that they have a faster convergence rate, $\mathcal{O}(\epsilon^{-2})$. Moreover, EGDA converges much faster than Smoothed-GDA.

## 6 Conclusions and Future Work

In this paper, we proposed a new single-loop accelerated algorithm for solving various constrained NC-NC minimax problems. In the proposed algorithm, we designed a novel extra-gradient difference scheme for dual updates. Moreover, we provided the convergence guarantees for the proposed algorithm, and the theoretical results show that to find an $\epsilon$-stationary point of $f$, the proposed algorithm obtains the complexity bound $\mathcal{O}(\epsilon^{-2})$ for constrained NC-NC problems. That is, this is the first time that the proposed algorithm attains the complexity bound $\mathcal{O}(\epsilon^{-2})$ in constrained NC-NC settings (including constrained NC-C and C-NC). For NC-C problems, we provided the theoretical guarantees of the proposed algorithm under the stationarity of $\phi$, which show that our algorithm improves the complexity bound from $\widetilde{\mathcal{O}}(\epsilon^{-3})$ to $\mathcal{O}(\epsilon^{-2})$. Experimental results also verified the theoretical results of the proposed algorithm, which have the factors of $\epsilon^{-1}$ and $\epsilon^{-2}$ faster than existing algorithms, respectively. For further work, we will extend our directly accelerating algorithm to stochastic, non-smooth, nonconvex-nonconcave and federated learning settings as in [47, 11, 4, 55, 45, 21, 37].

## Acknowledgments

We want to thank the anonymous reviewers for their valuable suggestions and comments. This work was supported by the National Key R&D Program of China (No. 2022ZD0160302), the National Natural Science Foundation of China (Nos. 6227071567, 61976164, 62276004 and 61836009), the National Science Basic Research Plan in Shaanxi Province of China (No. 2022GY-061), the major key project of PCL, China (No. PCL2021A12), and Qualcomm.

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
