## Appendix A: Basic Properties

Before giving our theoretical analysis, we first present the following basic properties.

**Property 2.** *Given any $a, b, c$, then we have*

$$\langle a - b, \ a - c \rangle = \frac{1}{2} \left( \|a - b\|^2 + \|a - c\|^2 - \|b - c\|^2 \right).$$

**Property 3.** *(Theorem 2.1.5 in [42]). If $f : \mathbb{R}^d \to \mathbb{R}$ is L-smooth, then for all $x, y \in \mathbb{R}^d$,*

$$\left| f(y) - f(x) - \nabla f(x)^T (y - x) \right| \leq \frac{L}{2} \|y - x\|^2.$$

**Property 4.** *[Lemma A.1 in [25]] Let $\phi_{1/2L}(\widehat{x}) := \min_x \{ \phi(x) + L\|x - \widehat{x}\|^2 \}$ and $x^*(\widehat{x}) = \arg\min_x \{ \phi(x) + L\|x - \widehat{x}\|^2 \}$. If $f : \mathbb{R}^d \to \mathbb{R}$ is L-smooth, and $\mathcal{Y}$ is bounded, then*

- *1. $\phi_{1/2L}$ is L-smooth with $\nabla\phi_{1/2L}(\widehat{x}) = 2L(\widehat{x} - x^*(\widehat{x}))$.*

- *2. $\phi_{1/2L}(x_1) - \phi_{1/2L}(x_2) - (x_1 - x_2)^T \nabla\phi_{1/2L}(x_2) \leq \frac{L}{2} \|x_1 - x_2\|^2$ for any $x_1, x_2 \in \mathbb{R}^m$.*

## Appendix B: Key Property

**Property 1.** *(**Quasi-Cocoercivity**) Let $u_{t+1/2}$ be updated in Eq. (3), then*

$$\left\langle \nabla_y f(x_t, u_{t-1/2}) - \nabla_y f(x_t, y_{t-1}), u_{t+1/2} - y_t \right\rangle = \beta \|\nabla_y f(x_t, u_{t-1/2}) - \nabla_y f(x_t, y_{t-1})\|^2. \quad (7)$$

*Proof.* By using the update rule $u_{t+1/2}$ in Eq. (3), that is, $u_{t+1/2} = y_t + \beta \left[ \nabla_y f(x_t, u_{t-1/2}) - \nabla_y f(x_t, y_{t-1}) \right]$, we get

$$\left\langle \nabla_y f(x_t, u_{t-1/2}) - \nabla_y f(x_t, y_{t-1}), u_{t+1/2} - y_t \right\rangle$$
$$= \beta \left\langle \nabla_y f(x_t, u_{t-1/2}) - \nabla_y f(x_t, y_{t-1}), \nabla_y f(x_t, u_{t-1/2}) - \nabla_y f(x_t, y_{t-1}) \right\rangle$$
$$= \beta \|\nabla_y f(x_t, u_{t-1/2}) - \nabla_y f(x_t, y_{t-1})\|^2.$$

This completes the proof. $\qquad\square$

## Appendix C: Proof of Lemma 1

Before proving Lemma 1, we first present and prove the following two propositions for the upper bounds of the primal variable and dual variable.

### Proof of Proposition 1

*Proof.* Using the optimality condition for the gradient descent step in Eq. (2) with respect to the primal variable $x \in \mathcal{X}$, we have

$$\left\langle \nabla_x f(x_t, y_t) + \frac{1}{\eta_x}(x_{t+1} - x_t), \ x_t - x_{t+1} \right\rangle \geq 0. \quad (8)$$

That is,

$$\langle \nabla_x f(x_t, y_t), x_{t+1} - x_t \rangle \leq -\frac{1}{\eta_x} \|x_{t+1} - x_t\|^2. \tag{9}$$

Since $f$ is $L$-smooth, then

$$
\begin{aligned}
f(x_{t+1}, u_{t+1/2}) \leq & f(x_t, y_t) + \langle \nabla_x f(x_t, y_t),\ x_{t+1} - x_t \rangle + \frac{L}{2} \|x_{t+1} - x_t\|^2 \\
& + \langle \nabla_y f(x_t, y_t),\ u_{t+1/2} - y_t \rangle + \frac{L}{2} \|u_{t+1/2} - y_t\|^2 \\
= & f(x_t, y_t) + \langle \nabla_x f(x_t, y_t),\ x_{t+1} - x_t \rangle + \frac{L}{2} \|x_{t+1} - x_t\|^2 \\
& + \langle \nabla_y f(x_t, y_{t-1}),\ u_{t+1/2} - y_t \rangle + \frac{L}{2} \|u_{t+1/2} - y_t\|^2 \\
& + \langle \nabla_y f(x_t, y_t) - \nabla_y f(x_t, y_{t-1}),\ u_{t+1/2} - y_t \rangle \\
\leq & f(x_t, y_t) - \left( \frac{1}{\eta_x} - \frac{L}{2} \right) \|x_{t+1} - x_t\|^2 \\
& + \langle \nabla_y f(x_t, y_{t-1}),\ u_{t+1/2} - y_t \rangle \\
& + L\|u_{t+1/2} - y_t\|^2 + \frac{1}{2L} \|\nabla_y f(x_t, y_t) - \nabla_y f(x_t, y_{t-1})\|^2 \\
\leq & f(x_t, y_t) - \left( \frac{1}{\eta_x} - \frac{L}{2} \right) \|x_{t+1} - x_t\|^2 \\
& + \langle \nabla_y f(x_t, y_{t-1}),\ u_{t+1/2} - y_t \rangle + L\|u_{t+1/2} - y_t\|^2 + L\|y_t - y_{t-1}\|^2,
\end{aligned}
\tag{10}
$$

where the second inequality holds due to the optimal condition in Eq. (9), and the last inequality holds due to the smoothness of $f$, i.e., $\|\nabla_y f(x_t, y_t) - \nabla_y f(x_t, y_{t-1})\| \leq L\|y_t - y_{t-1}\|$.

With $\eta_x < \frac{1}{L}$, and combining (9) and (10), we have

$$
\begin{aligned}
& f(x_{t+1}, u_{t+1/2}) - f(x_t, y_t) \\
\leq & - \left( \frac{1}{\eta_x} - \frac{L}{2} \right) \|x_{t+1} - x_t\|^2 + L\|u_{t+1/2} - y_t\|^2 + L\|y_t - y_{t-1}\|^2 + \underbrace{\langle \nabla_y f(x_t, y_{t-1}),\ u_{t+1/2} - y_t \rangle}_{A_1}.
\end{aligned}
\tag{11}
$$

This completes the proof. $\qquad\square$

**Proof of Proposition 2**

According to the update rule $y_{t+1} = \tau y_t + (1 - \tau) u_{t+1}$, we have

$$
\begin{aligned}
& y_{t+1} - u_{t+1} = \tau(y_t - u_{t+1}), y_{t+1} - y_t = (1 - \tau)(u_{t+1} - y_t) \\
& u_{t+1/2} - y_t = \beta(\nabla_y f(x_t, u_{t-1/2}) - \nabla_y f(x_t, y_{t-1})).
\end{aligned}
\tag{12}
$$

*Proof.* For given $x$, $f$ is $L$-smooth with respect to the dual variable $y$, and then we have

$$
\begin{aligned}
& f(x_{t+1}, y_{t+1}) - f(x_{t+1}, u_{t+1/2}) \\
\leq & \langle \nabla_y f(x_{t+1}, u_{t+1/2}),\ y_{t+1} - u_{t+1/2} \rangle + \frac{L}{2} \|y_{t+1} - u_{t+1/2}\|^2 \\
= & \langle \nabla_y f(x_t, u_{t-1/2}),\ y_{t+1} - u_{t+1} \rangle + \langle \nabla_y f(x_{t+1}, u_{t+1/2}),\ u_{t+1} - u_{t+1/2} \rangle \\
& + \frac{L}{2} \|y_{t+1} - u_{t+1/2}\|^2 + \langle \nabla_y f(x_{t+1}, u_{t+1/2}) - \nabla_y f(x_t, u_{t-1/2}),\ y_{t+1} - u_{t+1} \rangle \\
= & \tau \langle \nabla_y f(x_t, u_{t-1/2}),\ y_t - u_{t+1} \rangle + \langle \nabla_y f(x_t, u_{t-1/2}),\ u_{t+1} - u_{t+1/2} \rangle \\
& + \langle \nabla_y f(x_{t+1}, u_{t+1/2}) - \nabla_y f(x_t, u_{t-1/2}),\ y_{t+1} - u_{t+1} \rangle + \frac{L}{2} \|y_{t+1} - u_{t+1/2}\|^2 \\
& + \langle \nabla_y f(x_{t+1}, u_{t+1/2}) - \nabla_y f(x_t, u_{t-1/2}),\ u_{t+1} - u_{t+1/2} \rangle \\
= & \tau \underbrace{\langle \nabla_y f(x_t, u_{t-1/2}),\ y_t - u_{t+1} \rangle}_{A_2} + \underbrace{\langle \nabla_y f(x_t, u_{t-1/2}),\ u_{t+1} - u_{t+1/2} \rangle}_{A_3} + a_t,
\end{aligned}
\tag{13}
$$

where the last inequality holds due to $a_t := 2L\|x_{t+1} - x_t\|^2 + 6L\beta^2\|\nabla_y f(x_t, u_{t-1/2}) - \nabla_y f(x_t, y_{t-1})\|^2 + 8L(1 - \tau)^2\|u_t - y_{t-1}\|^2 + 8L\beta^2\|\nabla_y f(x_{t-1}, u_{t-3/2}) - \nabla_y f(x_{t-1}, y_{t-2})\|^2 + 3L\|u_{t+1} - y_t\|^2$ and

$L$-smoothness of $f$, i.e., $\|\nabla_y f(x_1, y_1) - \nabla_y f(x_2, y_2)\| \le L\|x_1 - x_2\| + L\|y_1 - y_2\|$ and

$$
\begin{aligned}
a_t =& \langle \nabla_y f(x_{t+1}, u_{t+1/2}) - \nabla_y f(x_t, u_{t-1/2}),\ y_{t+1} - u_{t+1} \rangle + \frac{L}{2}\|y_{t+1} - u_{t+1/2}\|^2 \\
& + \langle \nabla_y f(x_{t+1}, u_{t+1/2}) - \nabla_y f(x_t, u_{t-1/2}),\ u_{t+1} - u_{t+1/2} \rangle \\
\le& \frac{1}{L}\|\nabla_y f(x_{t+1}, u_{t+1/2}) - \nabla_y f(x_t, u_{t-1/2})\|^2 + \frac{L}{2}\|u_{t+1} - u_{t+1/2}\|^2 + \frac{L}{2}\|y_{t+1} - u_{t+1/2}\|^2 \\
& + \frac{L\tau^2}{2}\|u_{t+1} - y_t\|^2 \\
\le& 2L\|x_{t+1} - x_t\|^2 + 2L\|u_{t+1/2} - u_{t-1/2}\|^2 + \frac{L}{2}\|u_{t+1} - u_{t+1/2}\|^2 + \frac{L}{2}\|y_{t+1} - u_{t+1/2}\|^2 + \frac{L\tau^2}{2}\|u_{t+1} - y_t\|^2 \\
\le& 2L\|x_{t+1} - x_t\|^2 + 4L\|u_{t+1/2} - y_t\|^2 + 8L\|y_t - y_{t-1}\|^2 + 8L\|y_{k-1} - u_{t-1/2}\|^2 + \frac{L\tau^2}{2}\|u_{t+1} - y_t\|^2 \\
& + L\|u_{t+1} - y_t\|^2 + L\|y_t - u_{t+1/2}\|^2 + L\|y_{t+1} - y_t\|^2 + L\|y_t - u_{t+1/2}\|^2 \\
\le& 2L\|x_{t+1} - x_t\|^2 + 6L\|u_{t+1/2} - y_t\|^2 + 8L(1-\tau)^2\|u_t - y_{t-1}\|^2 + 8L\|y_{k-1} - u_{t-1/2}\|^2 \\
& + 2L\|u_{t+1} - y_t\|^2 + L(1-\tau)^2\|u_{t+1} - y_t\|^2 \\
=& 2L\|x_{t+1} - x_t\|^2 + 6L\beta^2\|\nabla_y f(x_t, u_{t-1/2}) - \nabla_y f(x_t, y_{t-1})\|^2 + 8L(1-\tau)^2\|u_t - y_{t-1}\|^2 \\
& + 8L\beta^2\|\nabla_y f(x_{t-1}, u_{t-3/2}) - \nabla_y f(x_{t-1}, y_{t-2})\|^2 + 3L\|u_{t+1} - y_t\|^2.
\end{aligned}
\tag{14}
$$

This completes the proof. $\qquad\square$

**Proof of Lemma 1**

*Proof.* We first estimate the bound of $A_1 + A_3$, where $A_1$ is given in (11) and $A_3$ is given in (13)

$$
\begin{aligned}
& A_1 + A_3 \\
=& \ \langle \nabla_y f(x_t, y_{t-1}),\ u_{t+1/2} - y_t \rangle + \langle \nabla_y f(x_t, u_{t-1/2}),\ u_{t+1} - u_{t+1/2} \rangle \\
=& \ \langle \nabla_y f(x_t, y_{t-1}) - \nabla_y f(x_t, u_{t-1/2}),\ u_{t+1/2} - y_t \rangle + \langle \nabla_y f(x_t, u_{t-1/2}),\ u_{t+1} - y_t \rangle \\
=& \ -\beta\|\nabla_y f(x_t, u_{t-1/2}) - \nabla_y f(x_t, y_{t-1})\|^2 + \langle \nabla_y f(x_t, u_{t-1/2}),\ u_{t+1} - y_t \rangle \\
\le& \ -\beta\|\nabla_y f(x_t, u_{t-1/2}) - \nabla_y f(x_t, y_{t-1})\|^2 + \frac{\eta_y^t}{2}\|\nabla_y f(x_t, u_{t-1/2})\|^2 + \frac{1}{2\eta_y^t}\|u_{t+1} - y_t\|^2 \\
=& \ -\beta\|\nabla_y f(x_t, u_{t-1/2}) - \nabla_y f(x_t, y_{t-1})\|^2 + \frac{\eta_y^t}{2}\|\nabla_y f(x_t, u_{t-1/2})\|^2 \\
& + \frac{1}{2\eta_y^t}\|\mathcal{P}_{\mathcal{Y}}\left(y_t + \eta_y^t \nabla_y f(x_t, u_{t-1/2})\right) - \mathcal{P}_{\mathcal{Y}}(y_t)\|^2 \\
\le& \ -\beta\|\nabla_y f(x_t, u_{t-1/2}) - \nabla_y f(x_t, y_{t-1})\|^2 + \eta_y^t\|\nabla_y f(x_t, u_{t-1/2})\|^2 \\
\le& \ -\beta\|\nabla_y f(x_t, u_{t-1/2}) - \nabla_y f(x_t, y_{t-1})\|^2 + \frac{\beta\|\nabla_y f(x_t, u_{t-1/2}) - \nabla_y f(x_t, y_{t-1})\|^2}{2\|\nabla_y f(x_t, u_{t-1/2})\|^2}\|\nabla_y f(x_t, u_{t-1/2})\|^2 \\
=& \ -\frac{\beta}{2}\|\nabla_y f(x_t, u_{t-1/2}) - \nabla_y f(x_t, y_{t-1})\|^2.
\end{aligned}
\tag{15}
$$

where the third equality holds follows from our Quasi-Cocoercivity in Property 1, the second inequality holds due to the non-expensive property of projection operator, that is, $\|\mathcal{P}_{\mathcal{Y}}\left(y_t + \eta_y^t \nabla_y f(x_t, u_{t-1/2})\right) - \mathcal{P}_{\mathcal{Y}}(y_t)\|^2 \le (\eta_y^t)^2\|\nabla_y f(x_t, u_{t-1/2})\|^2$, and the third inequality holds due to the stepsize $\eta_y^t = \min\{\frac{\beta\|\nabla_y f(x_t, u_{t-1/2}) - \nabla_y f(x_t, y_{t-1})\|^2}{2\|\nabla_y f(x_t, u_{t-1/2})\|^2}, \frac{1}{28L}\}$.

Furthermore, the optimality condition in (16) for the update of $u_{t+1}$ in (4) implies that $\forall u \in \mathcal{Y}$ and $\forall k \ge 1$

$$
\left\langle \nabla_y f(x_t, u_{t-1/2}) - \frac{1}{\eta_y^t}(u_{t+1} - y_t),\ u - u_{t+1} \right\rangle \le 0.
\tag{16}
$$

Next, we first bound $A_2$ in (13). Using the optimal condition in (16), with $u = y_t$, we have

$$A_2 = \langle \tau \nabla_y f(x_t, u_{t-1/2}), \ y_t - u_{t+1} \rangle$$
$$\leq -\frac{\tau}{\eta_y^t} \|u_{t+1} - y_t\|^2. \tag{17}$$

We recall the results of Propositions 1 and 2 as follows:

$$f(x_{t+1}, u_{t+1/2}) - f(x_t, y_t)$$
$$\leq -\left(\frac{1}{\eta_x} - \frac{L}{2}\right) \|x_{t+1} - x_t\|^2 + L\|u_{t+1/2} - y_t\|^2 + L\|y_t - y_{t-1}\|^2 + \underbrace{\langle \nabla_y f(x_t, y_{t-1}), \ u_{t+1/2} - y_t \rangle}_{A_1}.$$

and

$$f(x_{t+1}, y_{t+1}) - f(x_{t+1}, u_{t+1/2}) \leq A_2 + A_3 + a_t.$$

By the above three inequalities and the update rules in (12), we have

$$f(x_{t+1}, y_{t+1}) - f(x_t, y_t)$$
$$\leq A_1 + A_2 + A_3 + a_t$$
$$\quad -\left(\frac{1}{\eta_x} - \frac{L}{2}\right) \|x_{t+1} - x_t\|^2 + L\|u_{t+1/2} - y_t\|^2 + L\|y_t - y_{t-1}\|^2$$
$$\leq -\frac{\tau}{\eta_y^t} \|u_{t+1} - y_t\|^2 - \frac{\beta}{2} \|\nabla_y f(x_t, u_{t-1/2}) - \nabla_y f(x_t, y_{t-1})\|^2+$$
$$\quad -\left(\frac{1}{\eta_x} - \frac{L}{2}\right) \|x_{t+1} - x_t\|^2 + L\beta^2 \|\nabla_y f(x_t, u_{t-1/2}) - \nabla_y f(x_t, y_{t-1})\|^2 + L(1-\tau)^2 \|u_t - y_{t-1}\|^2 + a_t. \tag{18}$$

With $a_t := 2L\|x_{t+1} - x_t\|^2 + 6L\beta^2 \|\nabla_y f(x_t, u_{t-1/2}) - \nabla_y f(x_t, y_{t-1})\|^2 + 8L(1-\tau)^2 \|u_t - y_{t-1}\|^2 + 8L\beta^2 \|\nabla_y f(x_{t-1}, u_{t-3/2}) - \nabla_y f(x_{t-1}, y_{t-2})\|^2 + 3L\|u_{t+1} - y_t\|^2$ and the update rules in (12), we have

$$f(x_{t+1}, y_{t+1}) - f(x_t, y_t)$$
$$\leq -\frac{\tau}{\eta_y^t} \|u_{t+1} - y_t\|^2 - \frac{\beta}{2} \|\nabla_y f(x_t, u_{t-1/2}) - \nabla_y f(x_t, y_{t-1})\|^2$$
$$\quad -\left(\frac{1}{\eta_x} - 2L\right) \|x_{t+1} - x_t\|^2 + 7L\beta^2 \|\nabla_y f(x_t, u_{t-1/2}) - \nabla_y f(x_t, y_{t-1})\|^2 + 9L\|u_t - y_{t-1}\|^2$$
$$\quad + 8L\beta^2 \|\nabla_y f(x_{t-1}, u_{t-3/2}) - \nabla_y f(x_{t-1}, y_{t-2})\|^2 + 3L\|u_{t+1} - y_t\|^2. \tag{19}$$

Furthermore, using the definition of $G_t := f(x_t, y_t) + 9L\|u_t - y_{t-1}\|^2 + 8L\beta^2 \|\nabla_y f(x_{t-1}, u_{t-3/2}) - \nabla_y f(x_{t-1}, y_{t-2})\|^2$ and the update rules in (12) with $\eta_x \leq \frac{1}{4L}, \eta_y^t \leq \frac{1}{28L}$ and $\tau > 1/2$, we have

$$G_t - G_{t+1}$$
$$\geq \left(\frac{\tau}{\eta_y^t} - 7L\right) \|u_{t+1} - y_t\|^2 + \left(\frac{\beta}{2} - 15L\beta^2\right) \|\nabla_y f(x_t, u_{t-1/2}) - \nabla_y f(x_t, y_{t-1})\|^2$$
$$\quad + \frac{1}{2\eta_x} \|x_{t+1} - x_t\|^2 \tag{20}$$
$$\geq \frac{1}{4\eta_y^t} \|u_{t+1} - y_t\|^2 + \frac{\beta - 30L\beta^2}{2} \|\nabla_y f(x_t, u_{t-1/2}) - \nabla_y f(x_t, y_{t-1})\|^2 + \frac{1}{2\eta_x} \|x_{t+1} - x_t\|^2$$
$$\geq \frac{1}{4\eta_y^t} \|u_{t+1} - y_t\|^2 + \frac{\beta - 30L\beta^2}{2} \|\nabla_y f(x_t, u_{t-1/2}) - \nabla_y f(x_t, y_{t-1})\|^2 + \frac{1}{2\eta_x} \|x_{t+1} - x_t\|^2.$$

By the above analysis, we have

$$G_0 - G_T$$
$$= \sum_{t=0}^{T-1} (G_t - G_{t+1}) \tag{21}$$
$$\geq \sum_{t=0}^{T-1} \left(\frac{1}{4\eta_y^t} \|u_{t+1} - y_t\|^2 + \frac{\beta - 30L\beta^2}{2} \|\nabla_y f(x_t, u_{t-1/2}) - \nabla_y f(x_t, y_{t-1})\|^2 + \frac{1}{2\eta_x} \|x_{t+1} - x_t\|^2\right)$$

This completes the proof. $\qquad\qquad\square$

# Appendix D: Proofs of Theorems 1 and 2

## Proof of Theorem 1

*Proof.* Using the definition of the convergence criterion and the update rules in Algorithm 1, we define $\pi_t$ as $\pi_t := \pi(x_t, y_t)$, where $\pi(x_t, y_t)$ is defined in (6) in Definition 1. Using the update rules of the primal variable $x$ and dual variable $y$ in Algorithm 1, non-expensive property of $\mathcal{P}_{\mathcal{X}}$ and smoothness in Assumption 1, we have

$$
\begin{aligned}
\|(\pi_t)_x\| &= \frac{1}{\eta_x} \|x_t - \mathcal{P}_{\mathcal{X}}(x_t - \eta_x \nabla_x f(x_t, y_t))\| \\
&= \frac{1}{\eta_x} \|\mathcal{P}_{\mathcal{X}}(x_t) - \mathcal{P}_{\mathcal{X}}(x_t - \eta_x \nabla_x f(x_t, y_t))\| \\
&\leq \|\nabla_x f(x_t, y_t)\|
\end{aligned}
\tag{22}
$$

On the other hand, using the update rule of $u_{t+1}$ in Algorithm 1, non-expensive property of $\mathcal{P}_{\mathcal{Y}}$ and smoothness in Assumption 1,

$$
\begin{aligned}
\|(\pi_t)_y\| &= \frac{1}{\eta_y^t} \left\|y_t - \mathcal{P}_{\mathcal{Y}}\left(y_t + \eta_y^t \nabla_y f(x_t, y_t)\right)\right\| \\
&= \frac{1}{\eta_y^t} \left\|y_t - u_{t+1} + u_{t+1} - \mathcal{P}_{\mathcal{Y}}\left(y_t + \eta_y^t \nabla_y f(x_t, y_t)\right)\right\| \\
&= \frac{1}{\eta_y^t} \left\|y_t - u_{t+1} + \mathcal{P}_{\mathcal{Y}}\left(y_t + \eta_y^t \nabla_y f(x_t, u_{t-1/2})\right) - \mathcal{P}_{\mathcal{Y}}\left(y_t + \eta_y^t \nabla_y f(x_t, y_t)\right)\right\| \\
&\leq \frac{1}{\eta_y^t} \|y_t - u_{t+1}\| + \|\nabla_y f(x_t, y_t) - \nabla_y f(x_t, y_{t-1})\| + \left\|\nabla_y f(x_t, y_{t-1}) - \nabla_y f(x_t, u_{t-1/2})\right\| \\
&\leq \frac{1}{\eta_y^t} \|y_t - u_{t+1}\| + (1-\tau)L \|u_t - y_{t-1}\| + \left\|\nabla_y f(x_t, y_{t-1}) - \nabla_y f(x_t, u_{t-1/2})\right\|
\end{aligned}
\tag{23}
$$

Using the above two results with $\eta_y^t = \min\{\frac{\beta\|\nabla_y f(x_t, u_{t-1/2}) - \nabla_y f(x_t, y_{t-1})\|^2}{2\|\nabla_y f(x_t, u_{t-1/2})\|^2}, \frac{1}{28L}, \eta_x\}$ and $\tau > 1/2$ then

$$
\begin{aligned}
\|\pi_t\|^2 &= \|(\pi_t)_x\|^2 + \|(\pi_t)_y\|^2 \\
&\leq \|\nabla_x f(x_t, y_t)\|^2 + 2(1-\tau)^2 \|u_t - y_{t-1}\|^2 + 2\left\|\nabla_y f(x_t, y_{t-1}) - \nabla_y f(x_t, u_{t-1/2})\right\|^2 \\
&= \|\nabla_x f(x_t, y_t)\|^2 + (1-\tau)(\eta_y^t)^2 \left\|\nabla_y f(x_{t-1}, u_{t-3/2})\right\|^2 + 2\left\|\nabla_y f(x_t, y_{t-1}) - \nabla_y f(x_t, u_{t-1/2})\right\|^2 \\
&\leq \|\nabla_x f(x_t, y_t)\|^2 + \frac{(1-\tau)\eta_y^t}{28L} \left\|\nabla_y f(x_{t-1}, u_{t-3/2})\right\|^2 + 2\left\|\nabla_y f(x_t, y_{t-1}) - \nabla_y f(x_t, u_{t-1/2})\right\|^2 \\
&\leq \|\nabla_x f(x_t, y_t)\|^2 + \frac{\eta_y^t}{56L} \left\|\nabla_y f(x_{t-1}, u_{t-3/2})\right\|^2 + 2\left\|\nabla_y f(x_t, y_{t-1}) - \nabla_y f(x_t, u_{t-1/2})\right\|^2.
\end{aligned}
\tag{24}
$$

The result in (21) can be rewritten as follows:

$$
\begin{aligned}
&G_0 - G_T \\
&= \sum_{t=0}^{T-1} (G_t - G_{t+1}) \\
&\geq \sum_{t=0}^{T-1} \left(\frac{1}{4\eta_y^t} \|u_{t+1} - y_t\|^2 + \frac{\beta - 30L\beta^2}{2} \|\nabla_y f(x_t, u_{t-1/2}) - \nabla_y f(x_t, y_{t-1})\|^2 + \frac{\eta_x}{2} \|\nabla_x f(x_t, y_t)\|^2\right).
\end{aligned}
\tag{25}
$$

Similar to HiBSA [27] and APG [44], the first iteration index $T = T_\epsilon$ such that $\|\pi(x_t, y_t)\| \leq \epsilon$ is defined as: $T_\epsilon := \min\{t \mid \|\pi(x_t, y_t)\| \leq \epsilon\}$. Summing up the above inequality for all the iterations $t = 0, 1, \cdots, T_\epsilon - 1$ with $\beta < \frac{1}{60L}$, we obtain

$$
\begin{aligned}
& G_0 - G_{T_\epsilon} \\
=& \sum_{t=0}^{T_\epsilon - 1} [G_t - G_{t+1}] \\
\geq& \sum_{t=0}^{T_\epsilon - 1} \left( \frac{1}{4\eta_y^t} \|u_{t+1} - y_t\|^2 + \frac{\beta}{4} \|\nabla_y f(x_t, u_{t-1/2}) - \nabla_y f(x_t, y_{t-1})\|^2 + \frac{\eta_x}{2} \|\nabla_x f(x_t, y_t)\|^2 \right).
\end{aligned}
\tag{26}
$$

With $\eta_x \leq \frac{1}{4L}$ and $\beta < \frac{1}{60L}$, we have

$$
\sum_{t=0}^{T_\epsilon - 1} \|\pi_t\|^2 \leq 160L \Big[ G_0 - G_{T_\epsilon} \Big].
\tag{27}
$$

Using the definition of $G_t$, the following result holds

$$
\begin{aligned}
G_0 - G_{T_\epsilon} &= G_0 - f(x_{T_\epsilon}, y_{T_\epsilon}) - 9L\|y_{T_\epsilon - 1} - u_{T_\epsilon}\|^2 - 8L\beta^2 \|\nabla_x f(x_{T_\epsilon - 1}, u_{T_\epsilon - 3/2}) - \nabla_y f(x_{T_\epsilon - 1}, y_{T_\epsilon - 2})\|^2 \\
&\leq G_0 - f(x_{T_\epsilon}, y_{T_\epsilon}) \\
&= G_0 - f(x_{T_\epsilon}, u_{T_\epsilon - 1}) + f(x_{T_\epsilon}, u_{T_\epsilon - 1}) - f(x_{T_\epsilon}, y_{T_\epsilon}) \\
&\leq G_0 - f(x_{T_\epsilon}, u_{T_\epsilon - 1}) + \frac{L}{2}\|y_{T_\epsilon}^* - y_{T_\epsilon}\|^2 \\
&\leq G_0 - \underline{G} + 2LD_{\mathcal{Y}}^2.
\end{aligned}
\tag{28}
$$

Using the two inequalities, and the definition of $c_1$ in Theorem 1, we get

$$
\sum_{t=0}^{T_\epsilon - 1} \|\pi_t\|^2 \leq 160L \left( G_0 - G_{T_\epsilon} + 2LD_{\mathcal{Y}}^2 \right).
\tag{29}
$$

Thus,

$$
\frac{1}{T_\epsilon} \sum_{t=0}^{T_\epsilon - 1} \|\pi_t\|^2 \leq \frac{160L}{T_\epsilon} (G_0 - \underline{G} + 2LD_{\mathcal{Y}}^2),
$$

which implies that $\epsilon^2 \geq 160L \frac{G_0 - \underline{G} + 2LD_{\mathcal{Y}}^2}{T_\epsilon}$ or equivalently, $T_\epsilon \geq 160L \frac{G_0 - \underline{G} + 2LD_{\mathcal{Y}}^2}{\epsilon^2}$. That is, the gradient complexity of Algorithm 1 to find an $\epsilon$-stationary point of $f$ is

$$
\mathcal{O}\left( \frac{G_0 - \underline{G} + 2LD_{\mathcal{Y}}^2}{\epsilon^2} \right).
$$

This completes the proof. $\qquad\qquad\qquad\qquad\qquad\qquad\qquad\qquad\qquad\qquad\qquad\qquad\qquad$ $\square$

## Proof of Theorem 2

*Proof.* Let $x^*(\widehat{x}) = \arg\min_{x \in \mathcal{X}} \{\phi(x) + L\|x - \widehat{x}\|^2\}$, $\phi(x) := \max_{y \in \mathcal{Y}} f(x, y)$ and $\phi_{1/2L}(\widehat{x}) := \min_{x \in \mathcal{X}} \{\phi(x) + L\|x - \widehat{x}\|^2\}$. Using the definitions of $\phi$ and $\phi_{1/2L}$, and using Property 4, we have

$$
\|\nabla \phi_{1/2L}(\widehat{x})\|^2 = 4L^2 \|x^*(\widehat{x}) - \widehat{x}\|^2.
\tag{30}
$$

Following Property 3 and Lemma 19 in [24], the following result holds with the definition of $\phi(x)$

$$
\begin{aligned}
& \max_{y \in \mathcal{Y}} f(\widehat{x}, y) - \max_{y \in \mathcal{Y}} f(x^*(\widehat{x}), y) - L\|x^*(\widehat{x}) - \widehat{x}\|^2 \\
=& \phi(\widehat{x}) - \phi(x^*(\widehat{x})) - L\|x^*(\widehat{x}) - \widehat{x}\|^2 \\
\geq& \frac{L}{4}\|x^*(\widehat{x}) - \widehat{x}\|^2.
\end{aligned}
\tag{31}
$$

Let $\Omega^*$ be the solution set of the problem $\max_{y\in\mathcal{Y}} f(\widehat{x}, y)$, and $\Omega^{**}$ be the solution set of the problem $\max_{y\in\mathcal{Y}} f(x^*(\widehat{x}), y)$. And let $y^*(\widehat{x}) \in \Omega^*$ and $y^*(x^*(\widehat{x})) \in \Omega^{**}$, respectively, we get

$$\max_{y\in\mathcal{Y}} f(\widehat{x}, y) - \max_{y\in\mathcal{Y}} f(x^*(\widehat{x}), y) - L\|x^*(\widehat{x}) - \widehat{x}\|^2$$

$$= f(\widehat{x}, y^*(\widehat{x})) - f(x^*(\widehat{x}), y^*(x^*(\widehat{x})) - L\|x^*(\widehat{x}) - \widehat{x}\|^2$$

$$= f(\widehat{x}, y^*(\widehat{x})) - f(\widehat{x}, \widehat{y}) + f(\widehat{x}, \widehat{y}) - f(x^*(\widehat{x}), \widehat{y}) + f(x^*(\widehat{x}), \widehat{y}) - f(x^*(\widehat{x}), y^*(x^*(\widehat{x}))) - L\|x^*(\widehat{x}) - \widehat{x}\|^2$$

$$\leq \langle \nabla_y f(\widehat{x}, \widehat{y}),\ y^*(\widehat{x}) - \widehat{y}\rangle + \langle \nabla_x f(\widehat{x}, \widehat{y}),\ \widehat{x} - x^*(\widehat{x})\rangle - \frac{L}{2}\|x^*(\widehat{x}) - \widehat{x}\|^2,$$

$$\tag{32}$$

where the last inequality holds due to the concavity of $f$ w.r.t. $y$ (i.e., $f(\widehat{x}, y^*) \leq f(\widehat{x}, \widehat{y}) + \langle \nabla_y f(\widehat{x}, \widehat{y}),\ y^* - \widehat{y}\rangle$) ; the smoothness of $f$ (i.e., $f(x^*(\widehat{x}), \widehat{y}) \leq f(\widehat{x}, \widehat{y}) + \langle \nabla_x f(\widehat{x}, \widehat{y}),\ x^*(\widehat{x}) - \widehat{x}\rangle + \frac{L}{2}\|x^*(\widehat{x}) - \widehat{x}\|^2$) and $f(x^*(\widehat{x}), \widehat{y}) - f(x^*(\widehat{x}), y^*(x^*(\widehat{x}))) \leq 0$ (i.e., $y^*(x^*(\widehat{x}))$ is a solution of $\max_{y\in\mathcal{Y}} f(x^*(\widehat{x}), y)$).

With $\widehat{x} = x_t, \widehat{y} = u_t, y^*(\widehat{x}) = y^*(x_t)$, we have

$$\frac{\eta_y^t}{16L}\|\nabla\phi_{1/2L}(\widehat{x})\|^2 = \frac{L\eta_y^t}{4}\|x^*(\widehat{x}) - \widehat{x}\|^2$$

$$\leq \eta_y^t\left[\max_{y\in\mathcal{Y}} f(x_t, y) - \max_{y\in\mathcal{Y}} f(x^*(x_t), y) - L\|x^*(x_t) - x_t\|^2\right]$$

$$\leq \eta_y^t\left[\langle\nabla_y f(x_t, u_t),\ y^*(x_t) - u_t\rangle + \langle\nabla_x f(x_{t-1}, y_{t-1}),\ x_t - x^*(x_t)\rangle - \frac{L}{2}\|x^*(x_t) - x_t\|^2\right]$$

$$+ \eta_y^t\left[\langle\nabla_x f(x_t, u_t) - \nabla_x f(x_{t-1}, y_{t-1}),\ x_t - x^*(x_t)\rangle\right]$$

$$\leq \eta_y^t\left[\langle\nabla_y f(x_t, u_{t-1/2}),\ y^*(x_t) - u_t\rangle + \langle\nabla_x f(x_{t-1}, y_{t-1}),\ x_t - x^*(x_t)\rangle - \frac{L}{2}\|x^*(x_t) - x_t\|^2\right]$$

$$+ \eta_y^t\left[\langle\nabla_y f(x_t, u_t) - \nabla_y f(x_t, u_{t-1/2}),\ y^*(x_t) - u_t\rangle + \langle\nabla_x f(x_t, u_{t-1/2}) - \nabla_x f(x_{t-1}, y_{t-1}),\ x_t - x^*(x_t)\rangle\right]$$

$$\overset{a}{\leq} 2\mathcal{D}_y\eta_y^t\|\nabla_y f(x_t, u_{t-1/2})\| + \frac{\eta_y^t}{\eta_x}\langle x_t - x_{t-1},\ x^*(x_t) - x_t\rangle - \frac{L\eta_y^t}{4}\|x^*(x_t) - x_t\|^2$$

$$+ 2\mathcal{D}_y\eta_y^t\|\nabla_y f(x_t, u_t) - \nabla_y f(x_t, u_{t-1/2})\| + 4L\eta_y^t\|x_t - x_{t-1}\|^2 + 4L\eta_y^t\|u_{t-1/2} - y_{t-1}\|^2$$

$$\overset{b}{\leq} \beta\mathcal{D}_y\|\nabla_y f(x_t, y_{t-1}) - \nabla_y f(x_t, u_{t-1/2})\|^2 + \frac{\eta_y^t}{L\eta_x^2}\|x_t - x_{t-1}\|^2 + \frac{L\eta_y^t}{4}\|x^*(x_t) - x_t\|^2 - \frac{L\eta_y^t}{4}\|x^*(x_t) - x_t\|^2$$

$$+ \beta\mathcal{D}_y\|\nabla_y f(x_t, u_{t-1/2})\|^2 + 4L\eta_y^t\|x_t - x_{t-1}\|^2 + 4L\eta_y^t\|\nabla_y f(x_t, y_{t-2}) - \nabla_y f(x_t, u_{t-3/2})\|^2,$$

$$= \beta\mathcal{D}_y\|\nabla_y f(x_t, y_{t-1}) - \nabla_y f(x_t, u_{t-1/2})\|^2 + \frac{\eta_y^t}{L\eta_x^2}\|x_t - x_{t-1}\|^2$$

$$+ \beta\mathcal{D}_y\|\nabla_y f(x_t, u_{t-1/2})\|^2 + 4L\eta_y^t\|x_t - x_{t-1}\|^2 + 4L\eta_y^t\|\nabla_y f(x_t, y_{t-2}) - \nabla_y f(x_t, u_{t-3/2})\|^2,$$

$$\tag{33}$$

where inequality $(a)$ holds due to the Cauchy-Schwarz inequality and inequality $(b)$ holds due to the stepsize $\eta_y^t = \min\{\frac{\beta\|\nabla_y f(x_t, u_{t-1/2}) - \nabla_y f(x_t, y_{t-1})\|^2}{2\|\nabla_y f(x_t, u_{t-1/2})\|^2},\ \frac{1}{28L},\ \frac{\beta\|\nabla_y f(x_t, u_{t-1/2}) - \nabla_y f(x_t, y_{t-1})\|^2}{2\|\nabla_y f(x_t, u_{t-1/2})\|},\ \frac{\beta\|\nabla_y f(x_t, u_{t-1/2})\|^2}{2\|\nabla_y f(x_t, u_t) - \nabla_y f(x_t, u_{t-1/2})\|}\}$, that is, $2\eta_y^t\|\nabla_y f(x_t, u_{t-1/2})\| \leq \beta\|\nabla_y f(x_t, u_{t-1/2}) - \nabla_y f(x_t, y_{t-1})\|^2$ and $2\eta_y^t\|\nabla_y f(x_t, u_t) - \nabla_y f(x_t, u_{t-1/2})\| \leq \beta\|\nabla_y f(x_t, u_{t-1/2})\|^2$.

Using Lemma 1 and by the similar derivation with Theorem 1, the complexity of Algorithm 1 to find an $\epsilon$-stationary point of $\phi$ is

$$\mathcal{O}\left(\frac{\mathcal{D}_{\mathcal{Y}}(G_0 - \underline{G} + 2L\mathcal{D}_{\mathcal{Y}}^2)}{\epsilon^2}\right).$$

This completes the proof.

$\square$