# OpenReview forum: "A Single-Loop Accelerated Extra-Gradient Difference Algorithm with Improved Complexity Bounds for Constrained Minimax Optimization"
_NeurIPS.cc/2023/Conference — NeurIPS 2023 oral_

### Official Review · Reviewer_Lyoc · 2023-06-28

**Soundness:** 4 excellent
**Presentation:** 4 excellent
**Contribution:** 4 excellent
**Rating:** 9
**Confidence:** 5

**Summary:**

Authors propose method of solving nonconvex-nonconcave saddle point problems with convergence rate O(eps^-2) by using gradient difference prediction and momentum acceleration to improve extragradient descent-ascent method. Proposed method is state-of-the-art in theory and leading in practice, including neural network learning with adversarial attacks task.

**Strengths:**

Quite elegant construction of the algorithm, which also allows one to obtain best-known convergence rate guarantees. Algorithms allows practitioners to address the most practically important setting of nonconvex-nonconcave problems efficiently using easy to implement algorithm which will surely replace analogous methods, judging from empirical study.

**Weaknesses:**

No significant weaknesses

**Questions:**

Authors could provide more extensive empirical study, because algorithm seems to be candidate for being widely-accepted in practice and it would be good to have more justifications of its efficiency.

**Limitations:**

Everything is okay

---

> ### Author Rebuttal · Authors · 2023-08-09
>
> **Q**: Authors could provide more extensive empirical study, because algorithm seems to be candidate for being widely-accepted in practice and it would be good to have more justifications of its efficiency.
>
> **A**: Thanks for your positive and valuable comments. To address your concerns, we have provided more extensive empirical study, and reported more experimental results, as shown in Figs. 3 and 4 in the PDF file (please see “global” response). That is, we have applied the proposed algorithm to solve some real-world applications, such as robust neural network in Figure 3 and Wasserstein GAN training in Figure 4. All the results show that the proposed algorithm performs much better than other algorithms such as GDA, MGDA and Smoothed-GDA, which also verified our theoretical results. All the results will be included in our final paper. The details are as follows:
>
> 1.	More Results for Robust Neural Network in Figure 3 in the PDF file (please see “global” response)
>
> Under adversarial attacks including $\ell_\infty$ -norm FGSM and PGD attacks, the test accuracies of all the algorithms including GDA, MGDA, Smoothed-GDA and our EGDA are reported in Fig.3, where the $\ell_\infty$ -norm perturbation level $\varepsilon$ varies from $0.0$ to $0.4$. Note that for EGDA, the parameter $\tau$ is set to $3/4$, and the parameters $\alpha$ and $\beta$ in Smoothed-GDA are set to $0.2$ and $0.8$ as in [51], respectively. And the number of iterations is set to 100 for all the algorithms. All the results show that Smoothed-GDA and EGDA significantly outperform GDA and MGDA in terms of accuracy, and our EGDA also performs better than other algorithms including Smoothed-GDA.
>
> 2.	More Results for Wasserstein GAN in Figure 4 in the PDF file (please see “global” response)
>
> Finally, we apply the stochastic version of the proposed EGDA algorithm to train Wasserstein GAN in [R3] on the MNIST dataset, and verify the effectiveness of our algorithm. Here the architectures of Wasserstein GAN (including its discriminator and generator) are set to be multi-layer perceptrons (MLP). The layer widths of the MLP in generator are 100, 128, 784, and the layer widths of the MLP in discriminator are 784, 128, 1. In addition, the batch size is set to 64, and the learning rate is 1e-4. Moreover, we compare our algorithm against one state-of-the-art method, Stochastic Gradient Descent Ascent (SGDA) by drawing their generated figures after 20k and 100k iterations, as shown in Fig.4. All the results show that our stochastic algorithm performs much better than SGDA and produces higher quality images, which shows the effectiveness of our algorithm.
>
> [R3] M. Arjovsky , et al., “Wasserstein generative adversarial networks,” ICML 2017.

---

> > ### Comment · Reviewer_Lyoc · 2023-08-19
> >
> > Dear authors, thank you for your work on the final version of your paper! The rebuttal has clarified my questions. I decided to keep my overall rating the same.

---

> > > ### Author Response · Authors · 2023-08-20
> > >
> > > We sincerely appreciate the reviewer for your positive and valuable comments. We are delighted to learn that our response effectively addressed your questions.

---

### Official Review · Reviewer_gZnc · 2023-07-05

**Soundness:** 3 good
**Presentation:** 4 excellent
**Contribution:** 4 excellent
**Rating:** 9
**Confidence:** 3

**Summary:**

The authors have designed a single-loop accelerated algorithm for constrained min-max optimization problems of the form $\min_{x\in X}\max_{y\in Y} f(x,y)$. The algorithm provably converges in an approximate local stationary point in three particular setting:
1. Non-convex non-concave min-max optimization, where the stationarity is measured for the function $f(x,y)$.
2. Convex non-concave min-max optimization, and non-convex concave min-max optimization, where the stationarity is measured for function $\phi(x)=\max_{y'\in Y} f(x,y')$.

The authors showed that their algorithm computes an $\epsilon$-stationary point, in $O(1/\epsilon^2)$ iterations. Finally, they experimentally verify their proposed algorithm.



The authors' rebuttal addressed my concerns, and their additional empirical evidence complemented their already compelling results. For these reasons, I decided to increase my score.

**Strengths:**

The design and analysis of algorithms for non-convex non-concave minimax optimization is a fundamental problem, and the convergent results are indeed compelling. Moreover, the authors get the state-of-the-art for convex non-concave, and concave non-convex for the merit function they consider. Furthermore the main paper is well-written and easy to follow, and the algorithm seems to combine several interesting ideas.

I verified the proofs of proposition 1 and 2, and the rest of the statement seems reasonable. I found the proofs of proposition 1 and 2 to be a bit dense, which made verifying them somewhat taxing.

**Weaknesses:**

Overall, I did not find some important weakness in the paper. As a suggestion, improving the readability and verifiability of the proofs could greatly benefit readers. Lastly, I came across a couple of typos:
- In line 242, I think "conference" was meant to be "convergence".
- When considering the proof of proposition 2 in the appendix, is $\widehat{u}_{t+1/2}$ identical to the one referred to in line 706? If so, clarifying this might prevent confusion.
- In line 3 in Algorithm 1, do you also need to initialize $y_{-1}$ for the first iteration of the algorithm?

**Questions:**

The recent work in [1], shows that the computation of an approximate stationary point is PPAD-complete when the action space of the two players is a joint. Notably, the findings in your paper seem to suggest a different outlook when the strategy space of the two players is a product space, which negates the hardness results. A comment from the authors addressing this observation would be greatly appreciated.

Furthermore, the proofs of the propositions and lemmas presented appear to be quite opaque, and verifying them poses a bit of a challenge. It would be truly beneficial if the authors could share some additional insights into the process behind the design and analysis of the algorithm. I think it would be good if the authors could provide a better commentary about the derivations of the proofs.

[1] "The Complexity of Constrained Min-Max Optimization" by Constantinos Daskalakis, Stratis Skoulakis, and Manolis Zampetakis

**Limitations:**

Any assumptions needed for the theorems to hold are listed. I do not believe this work can have negative societal impact.

---

> ### Author Rebuttal · Authors · 2023-08-09
>
> **Q**: The recent work in [1], shows that the computation of an approximate stationary point is PPAD-complete when the action space of the two players is a joint. Notably, the findings in your paper seem to suggest a different outlook when the strategy space of the two players is a product space, which negates the hardness results. A comment from the authors addressing this observation would be greatly appreciated.
> [1] "The Complexity of Constrained Min-Max Optimization" by Constantinos Daskalakis, Stratis Skoulakis, and Manolis Zampetakis.
>
> **A**:  Thanks for your positive and constructive comments. To address your concern, we will add this reference, and provide some discussions about [1] in our final paper.
>
> 1.	In [1], the authors studied a constrained nonconvex-nonconcave problem, $\min_x\max_y f(x,y), s.t., g(x,y)<=0$, where the funciton $g$ is a linear funciton so that the constraint set is a polytope. While the constrained sets in our paper are closed, convex and compact sets in domains $X$ and $Y$. That is, the constrained set in our paper is only a special case of [1]. Therefore, the problem used in our work is much simpler than that in [1].
>
> 2.	We design an extra-gradient difference iteration in our algorithm, which is similar to the forms in [43] (i.e., the difference of gradients) to achieve the approximation of negative curvature of a Hessian matrix. That is, it goes beyond a first-order method，while a first-order method is studied in [1]. The negative curvature method [43] can escape from saddle points for non-convex optimization problems. Thus, we think that our method may escape from the saddle point of lower level nonconcave problem w.r.t. $y$, which is very hard to first-order methods. That is, our method can find a "better" solution of lower level nonconcave problem w.r.t. $y$, which is an important reason to reduce the hardness.
>
> All the discussions will be included in the revised manuscript.
>
> **Q**: Furthermore, the proofs of the propositions and lemmas presented appear to be quite opaque, and verifying them poses a bit of a challenge. It would be truly beneficial if the authors could share some additional insights into the process behind the design and analysis of the algorithm. I think it would be good if the authors could provide a better commentary about the derivations of the proofs.
>
> **A**: To address your concern, we will provide more details about the derivations of the proofs, and add more detailed explanations for the process behind the design and analysis of the proposed algorithm in our final paper.
>
> **Q**: In line 242, I think "conference" was meant to be "convergence".
>
> **A**: To address your concern, we have corrected in the revised manuscript.
>
> **Q**: When considering the proof of proposition 2 in the appendix, is $\widehat{u}_{t+1/2}$ identical to the one referred to in line 706? If so, clarifying this might prevent confusion.
>
> **A**: Yes, $\widehat{u}_{t+1/2}$ is identical to the one referred to in line 706, and we have clarified this in the revised manuscript.
>
> **Q**:  In line 3 in Algorithm 1, do you also need to initialize $y_{-1}$ for the first iteration of the algorithm?
>
> **A**: Yes, we need to initialize $y_{-1}$, and have added such initialization in the revised manuscript.

---

> > ### Comment · Reviewer_gZnc · 2023-08-14
> >
> > I appreciate your response to my questions. Moreover, the empirical evidences presented are quite compelling. After reviewing the feedback and rebuttals from the other reviewers, I've decided to increase my score.

---

> > > ### Author Response · Authors · 2023-08-20
> > >
> > > We sincerely appreciate the reviewer for noticing that we have concrete contribution, and raising the score. We are delighted to learn that our response effectively addressed your questions.

---

### Official Review · Reviewer_RSU3 · 2023-07-07

**Soundness:** 3 good
**Presentation:** 3 good
**Contribution:** 3 good
**Rating:** 6
**Confidence:** 2

**Summary:**

This work proposes a single-loop extra-gradient difference acceleration algorithm to find an \epsilon-stationary point for constrained minimax optimization, which pushes forward the best complexity bounds of NC-NC, C-NC, NC-C problems to \mathcal{O}(\epsilon^{-2}). The proposed approach can deal with more general problems as it does not require monotone or structural assumption. Moreover, for the NC-C problem, the authors prove that the proposed method has better complexity bound under the stationarity of \phi. Experiments are conducted to validate the method empirically. The results show that it can achieve better convergence rate when comparing with the related methods.

**Strengths:**

1.	The theoretical contributions are significant. The method employs a novel prediction point scheme to obtain the quasi-cocoercivity property, which relaxes the assumption requirements. Additionally, the paper provides a thorough analysis of the convergence complexity bound, demonstrating its superiority over the current state-of-the-art approaches.

2.	The paper is well organized and easy to follow. The logical flow of ideas is well-structured, enhancing the overall readability and comprehension of the presented contents.

3.	Empirical studies are conducted to validate the method in both synthetic and real tasks.


**Weaknesses:**

There are some possible limitations where the paper could be further improved.

1.	I suggest the authors to undertake additional analysis of the algorithm's time complexities, both theoretically and empirically. This deeper exploration would provide valuable insights, particularly for potential industrial applications.

2.	The absence of experiments conducted on the C-NC problem should be explained within the paper.

3.	There are some empirical evidences that seem to be inconsistent with the theoretical result, for example, FEG has a fast theoretical rate but is less effective in practice; GDA may not converge to stationary points. The authors may explain more about these in the paper.


**Questions:**

please see above.

**Limitations:**

no negative societal impact

---

> ### Author Rebuttal · Authors · 2023-08-09
>
> **Q**:  I suggest the authors to undertake additional analysis of the algorithm's time complexities, both theoretically and empirically. This deeper exploration would provide valuable insights, particularly for potential industrial applications.
>
> **A**:   Thanks for your positive and valuable comments. To address your concern, we will add the time complexity of the proposed algorithm for solving  min-max problems in the final version. In our algorithm, three gradients need to be calculated for each iteration update, the time complexity for constrained NC-NC setting is O((m+n)\epsilon^{-2}), where m and n denote the dimensions of x and y, respectively. Moreover, we have conducted more empirical experiments and compared the performance of all the algorithms over the running time, as shown in Fig. 1 in the PDF file (please see “global” response). All the results show that the proposed algorithm converges significantly faster than other algorithms, as verified by theoretical and empirical analysis. All the results will be included in our final paper.
>
> **Q**: The absence of experiments conducted on the C-NC problem should be explained within the paper.
>
> **A**: To address your concern, we will add some discussions about the C-NC problem in the revised manuscript. In fact, there are few C-NC minimax problems in real-word applications, such as the convex-nonconcave zero sum game in [R1]. If necessary, we will add some experiments for solving such problem in the revised manuscript.
>
> [R1] G. Su, et al. Secrecy-oriented user association in ultra dense heterogeneous networks against strategically colluding adversaries. IET Commun., 2022.
>
> **Q**: There are some empirical evidences that seem to be inconsistent with the theoretical result, for example, FEG has a fast theoretical rate but is less effective in practice; GDA may not converge to stationary points. The authors may explain more about these in the paper.
>
> **A**:  To address your concern, we will make the following clarifications. In fact, FEG has a fast theoretical rate for the problem with an additional structured assumption. However, the NC-NC function used in Figure 1 in this paper does not satisfy the structured assumption. As a result, FEG has a slow convergence rate. In addition, we have conducted more experiments of FEG for the function in Figure 2 (please see “global” response), and reported empirical results in Fig. 2 in the PDF file. The results show that FEG converges much faster than GDA and EAG.

---

> > ### Author Response · Authors · 2023-08-20
> >
> > Thanks for your positive and valuable comments. We will improve the final version based on your comments.

---

### Official Review · Reviewer_VxsC · 2023-07-10

**Soundness:** 3 good
**Presentation:** 4 excellent
**Contribution:** 3 good
**Rating:** 8
**Confidence:** 4

**Summary:**

This paper discusses a new extra-gradient difference acceleration algorithm for solving constrained nonconvex-nonconcave minimax problems. The algorithm introduces a "quasi-cocoercivity property" and momentum acceleration to significantly improve the convergence rate in the constrained NC-NC setting. The algorithm attains a complexity of $O(\epsilon^{-2})$ for finding an $\epsilon$-stationary point of the function $f$, which outperforms the best-known complexity bounds. The paper also provides theoretical analysis and comparisons with existing algorithms.

**Strengths:**

As a person who works in minimax optimization, I can make a fair judgment of this work. This paper presents a novel extra-gradient difference acceleration algorithm for solving constrained nonconvex-nonconcave minimax problems, which improves the existing convergence rate and outperforms the best-known complexity bounds to $O(\epsilon^{-2})$. The paper also provides a comprehensive comparison with existing algorithms and a theoretical analysis of the algorithm's performance. I understand the "extra-gradient difference prediction" step as the key to the success of convergence rate improvements. In addition, I went through the proofs of Theorems 1 and 2 in detail. Overall, this paper provides valuable contributions to the field of minimax optimization and presents a promising algorithm for solving constrained NC-NC problems.

**Weaknesses:**

The paper assumes that the objective function satisfies certain structural assumptions, which may limit its practical applications. Also the writing style might not be as friendly for readers unfamiliar with the topic (I found it sufficiently clear though).

**Questions:**

---Can you explain more about the "quasi-cocoercivity" property and discuss how it improves the convergence rate in the constrained NC-NC setting? Is this an absolutely necessary property for the improved convergence rate to hold?

---Can your algorithm or its variants be applied to other minimax optimization problems beyond the constrained NC-NC setting?

---There do exist some typos. For example in Eq. (4) in Line 215, missing factor 2 in the denominator. Should be minor but the authors should check more and correct them.

**Limitations:**

This paper is purely theoretical and does not admit negative social impacts to my best knowledge.

---

> ### Author Rebuttal · Authors · 2023-08-09
>
> **Q**： Can you explain more about the "quasi-cocoercivity" property and discuss how it improves the convergence rate in the constrained NC-NC setting? Is this an absolutely necessary property for the improved convergence rate to hold?
>
> **A**:  Thanks for your positive and valuable comments. To address your concern, we will add some explanations about the "quasi-cocoercivity" property, and discuss how it improves the convergence rate in the constrained NC-NC setting in our final paper. From the perspective of theoretical analysis, we use the "quasi-cocoercivity" property to offset some residual terms produced by Propositions 1 and 2. As a result, we can obtain a descent sequence of the potential function, i.e., G_t. From the perspective of algorithmic intuition, the "quasi-cocoercivity" property is related to the extra-gradient difference iteration, which can improve the convergence rate. Thus, we think that it is a necessary property to improve the convergence rate.
>
> **Q**：Can your algorithm or its variants be applied to other minimax optimization problems beyond the constrained NC-NC setting?
>
> **A**:   To address your concern, we will make some discussions in the revised manuscript. The proposed algorithm can be extended to some problems beyond the constrained NC-NC setting. For example, for the robust neural network training problem, which does not require the compactness of the domain X (x is the parameter of the neural network), it goes beyond the constrained condition in the domain X. Furthermore, we can also extend the proposed algorithm to the stochastic setting to effectively solve large-scale problems. In addition, we also provide more experimental results  in Figs.3 and 4 in the PDF file for more real-world applications (please see “global” response).
>
> **Q**：There do exist some typos. For example in Eq. (4) in Line 215, missing factor 2 in the denominator. Should be minor but the authors should check more and correct them.
>
> **A**:   To address your concern, we have corrected these typos in the revised manuscript.

---

> > ### Comment · Reviewer_VxsC · 2023-08-18
> >
> > I appreciate the authors for the clarifications in their rebuttal, and I have raised my score from 7 to 8. I do encourage the authors to take more passes on their manuscript for typographical polishing/corrections before their final camera-ready submission.

---

> > > ### Author Response · Authors · 2023-08-20
> > >
> > > We sincerely appreciate the reviewer for raising the score. We must carefully corrected these typos on our manuscript before our final camera-ready submission.

---

### Author Rebuttal · Authors · 2023-08-09

Dear Reviewers and Area Chairs:

Thank you very much for the constructive comments. More experimental results in the PDF file, and the details are as follows:

**1**. We have conducted more empirical experiments and compared the performance of all the algorithms over the running time, as shown in Fig. 1 in the PDF file.

**2**. We have conducted more experiments of FEG for the function in Figure 2, and reported empirical results in Fig. 2 in the PDF file.

**3**. More Results for Robust Neural Network in Figure 3:

Under adversarial attacks including $\ell_\infty$-norm FGSM and PGD attacks, the test accuracies of all the algorithms including GDA, MGDA, Smoothed-GDA and our EGDA are reported in Fig.3, where the $\ell_\infty$-norm perturbation level $\varepsilon$ varies from $0.0$ to $0.4$. Note that for EGDA, the parameter $\tau$ is set to $3/4$, and the parameters $\alpha$ and $\beta$ in Smoothed-GDA are set to $0.2$ and $0.8$ as in [51], respectively. And the number of iterations is set to 100 for all the algorithms. All the results show that Smoothed-GDA and EGDA significantly outperform GDA and MGDA in terms of accuracy, and our EGDA also performs better than other algorithms including Smoothed-GDA.

**4**.	More Results for Wasserstein GAN in Figure 4

Finally, we apply the stochastic version of the proposed EGDA algorithm to train Wasserstein GAN in [R3] on the MNIST dataset, and verify the effectiveness of our algorithm. Here the architectures of Wasserstein GAN (including its discriminator and generator) are set to be multi-layer perceptrons (MLP). The layer widths of the MLP in generator are 100, 128, 784, and the layer widths of the MLP in discriminator are 784, 128, 1. In addition, the batch size is set to 64, and the learning rate is 1e-4. Moreover, we compare our algorithm against one state-of-the-art method, Stochastic Gradient Descent Ascent (SGDA) by drawing their generated figures after 20k and 100k iterations, as shown in Fig.4. All the results show that our stochastic algorithm performs much better than SGDA and produces higher quality images, which shows the effectiveness of our algorithm.

[R3] M. Arjovsky , et al., “Wasserstein generative adversarial networks,” ICML 2017.

---

### Decision · Program_Chairs · 2023-09-21

**Decision:**

Accept (oral)

**Comment:**

This paper introduces an extra-gradient difference acceleration algorithm for constrained nonconvex-nonconcave minimax problems. The algorithm improves upon the existing best convergence rate. Additionally, the paper offers an extensive comparison with current algorithms and delivers a thorough theoretical analysis. The whole review team believes that the paper makes significant contributions to the field of minimax optimization.